# See through the Dark: Learning Illumination-affined Representations for Nighttime Occupancy Prediction

**Yuan Wu**[1][*]   **Zhiqiang Yan**[2][*]   **Yigong Zhang**[3][†]   **Xiang Li**[3]   **Jian Yang**[1][†]

[1]PCA Lab[‡], Nanjing University of Science and Technology
[2]National University of Singapore    [3]Nankai University

## Abstract

Occupancy prediction aims to estimate the 3D spatial distribution of occupied regions along with their corresponding semantic labels. Existing vision-based methods perform well on daytime benchmarks but struggle in nighttime scenarios due to limited visibility and challenging lighting conditions. To address these challenges, we propose **LIAR**, a novel framework that **l**earns **i**llumination-**a**ffined **r**epresentations. LIAR first introduces Selective Low-light Image Enhancement (SLLIE), which leverages the illumination priors from daytime scenes to adaptively determine whether a nighttime image is genuinely dark or sufficiently well-lit, enabling more targeted global enhancement. Building on the illumination maps generated by SLLIE, LIAR further incorporates two illumination-aware components: 2D Illumination-guided Sampling (2D-IGS) and 3D Illumination-driven Projection (3D-IDP), to respectively tackle local underexposure and overexposure. Specifically, 2D-IGS modulates feature sampling positions according to illumination maps, assigning larger offsets to darker regions and smaller ones to brighter regions, thereby alleviating feature degradation in underexposed areas. Subsequently, 3D-IDP enhances semantic understanding in overexposed regions by constructing illumination intensity fields and supplying refined residual queries to the BEV context refinement process. Extensive experiments on both real and synthetic datasets demonstrate the superior performance of LIAR under challenging nighttime scenarios. The source code and pretrained models are available here.

## 1 Introduction

Understanding the 3D structure of the environment [15, 16, 52–55] is a core task in autonomous driving, as it allows vehicles to perceive their surroundings and make informed decisions. Recently, vision-based occupancy prediction [12, 18, 26, 31] has attracted growing interest due to its ability to estimate the spatial layout of occupied regions together with their semantic labels. Although existing methods perform well under daytime conditions, their performance drops significantly in nighttime scenes. The need for reliable perception in such low-light environments underscores the importance of advancing nighttime occupancy prediction. However, this task remains particularly challenging, as limited visibility and complex lighting conditions can severely degrade the quality of visual inputs.

As illustrated in Fig. 1(a), nighttime images not only exhibit low visibility but also contain both underexposed and overexposed regions [23, 39, 57]. Underexposure caused by insufficient illumination severely degrades visual features, whereas overexposure from artificial light sources, such as vehicle headlights and streetlamps, results in significant semantic deficiency in the saturated regions.

---

[*]Equal contribution.

[†]Corresponding authors.

[‡]PCA Lab, Key Lab of Intelligent Perception and Systems for High-Dimensional Information of Ministry of Education, School of Computer Science and Engineering, Nanjing University of Science and Technology.

39th Conference on Neural Information Processing Systems (NeurIPS 2025).

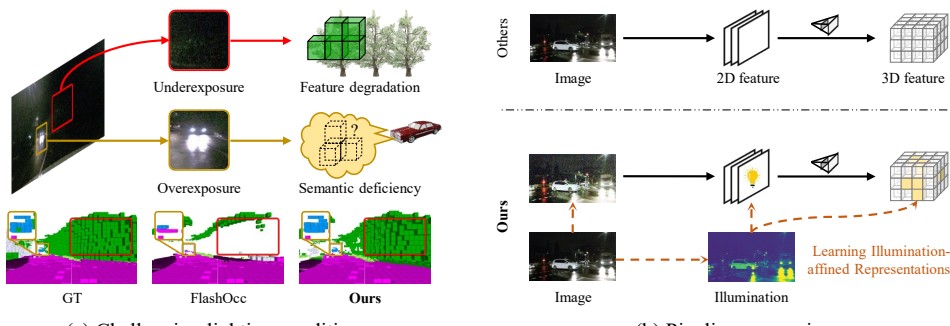

|                                    |                                 |
|:----------------------------------:|:-------------------------------:|
| (a) Challenging lighting conditions | (b) Pipeline comparison         |

Figure 1: (a) Nighttime images often suffer from both underexposure and overexposure, resulting in feature degradation and semantic deficiency. (b) Our approach learns illumination-aware representations to enhance both the fundamental 2D and 3D stages.

To address these challenges, we propose LIAR, a novel framework that learns illumination-affined representations to enhance and leverage nighttime visual information. LIAR introduces a Selective Low-light Image Enhancement (SLLIE) module, which first evaluates the global illumination of a nighttime image using priors derived from daytime scenes. This selective strategy ensures that only genuinely low-light images are enhanced, enabling the model to focus on challenging inputs while avoiding over-amplification of well-lit images that could otherwise lead to contrast degradation or overexposure. As shown in Fig. 1(b), building on the illumination maps produced by SLLIE, LIAR further incorporates two illumination-aware components, 2D Illumination-guided Sampling (2D-IGS) and 3D Illumination-driven Projection (3D-IDP), to respectively address underexposure and overexposure. In the 2D occupancy stage, 2D-IGS leverages illumination maps to adaptively generate feature sampling points, assigning larger offsets to darker regions and smaller offsets to brighter regions. This approach enables the model to effectively compensate for underexposed regions by aggregating semantic cues from adequately-exposed regions, thereby mitigating feature degradation and improving feature quality. In the 3D occupancy stage, 3D-IDP constructs 3D illumination intensity fields to refine the projection process by placing greater emphasis on overexposed regions. This targeted strategy mitigates the adverse effects of semantic deficiency caused by overexposure. Concurrently, these three modules enable LIAR to effectively address both global and local illumination challenges, improving occupancy prediction in complex nighttime environments.

In summary, our contributions are summarized as follows:

- To the best of our knowledge, we are the first to propose LIAR, a novel illumination-driven occupancy prediction framework tailored to the challenges of nighttime environments.
- We introduce three key components: SLLIE, 2D-IGS, and 3D-IDP. SLLIE enables global yet targeted image enhancement, while 2D-IGS and 3D-IDP respectively mitigate the negative effects of underexposure and overexposure in local regions.
- Extensive experiments on both real and synthetic datasets demonstrate the superiority of our approach, achieving up to a 7.9-point improvement over the second-best method. Source code and pretrained models are released for peer research.

## 2 Related Work

**Vision-based Occupancy Prediction.** Recently, vision-based 3D occupancy prediction has attracted growing interest, with studies exploring both supervised and unsupervised learning paradigms [12, 31, 32, 49, 64, 68]. Among supervised approaches, MonoScene [3] is a pioneering method for monocular occupancy prediction. BEVDet [11] adopts the Lift-Splat-Shoot (LSS) [34] for view transformation. BEVDet4D [9] further explores the temporal fusion strategy by fusing history frames. Building upon the BEVDet [11, 20, 21] series, FlashOcc [63] introduces a channel-to-height mechanism for memory-efficient occupancy prediction. In addition, several transformer-based methods have been proposed. For instance, BEVFormer [25] utilizes spatiotemporal transformers to construct BEV features. SparseOcc [27] takes the first step to explore the fully sparse architecture. For unsupervised methods [13, 66], SelfOcc [13] and OccNeRF [66] are two representative methods that employ volume

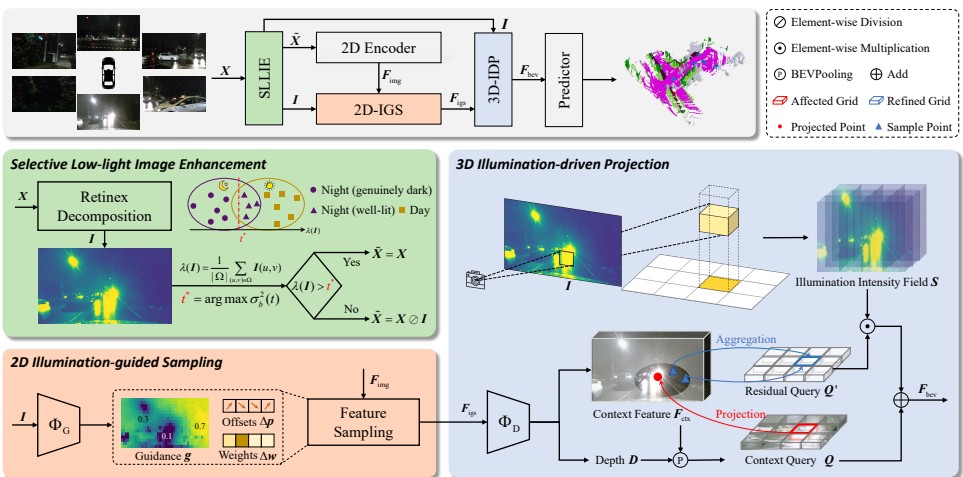

Figure 2: Pipeline of LIAR. The input $X$ is initially processed by SLLIE to generate an illumination map $I$ and an enhanced image $\tilde{X}$. Using $I$, the 2D-IGS module modulates feature sampling to enhance degraded features in underexposed regions, while the 3D-IDP module constructs 3D illumination intensity fields to compensate for deficient semantics in overexposed regions. Finally, a prediction head is deployed to output the occupancy.

rendering to generate self-supervised signals. Recently, Gaussian-based representations [6, 17, 42] have emerged as a powerful approach for 3D scene modeling. For example, GaussianTR [17] aligns Gaussian features with foundation models, achieving state-of-the-art zero-shot performance.

**Low-light Image Enhancement.** Low-light image enhancement [8, 23, 36, 41, 48, 57] aims to improve the brightness and contrast. Histogram Equalization [35], along with its variant CLHE [35], is a classical method for enhancing the global contrast of images. However, their effectiveness diminishes in the presence of significant image noise or non-uniform illumination. Retinex-based methods [36] offer an alternative strategy. Recently, network-based Retinex methods [2, 28, 36, 46] combine convolutional neural networks with Retinex theory to achieve improved accuracy. Nevertheless, in real-world nighttime driving scenarios, artificial light sources often cause spatially uneven illumination [23, 39]. To address this issue, LCDPNet [39] leverages local color distributions to correct illumination imbalance, while CSEC [23] models color distribution shifts to enhance image quality. Different from the aforementioned methods, we do not uniformly treat all nighttime images as requiring enhancement. Instead, we selectively enhance images that suffer from poor illumination, as indiscriminate enhancement may degrade the visual quality of already well-lit ones.

**View Transformation.** Transforming 2D image features into 3D space is a crucial step in 3D perception tasks [58–62]. Bottom-up methods [20, 21, 34] rely on depth estimation [43–45, 56] to project multi-view features into the BEV space, a paradigm first introduced by LSS [34]. However, the resulting BEV features are typically sparse, posing challenges for tasks that require dense spatial representations. In contrast, top-down transformer-based methods [22, 25, 47] generate dense BEV features, but their lack of explicit geometric constraints makes them vulnerable to occlusions and depth inconsistencies. Recent studies [24, 31, 37, 65] attempt to integrate the strengths of both paradigms. For instance, FB-OCC [24] leverages LSS-generated features to initialize transformer queries, while COTR [31] extends this strategy by employing compact occupancy representations to preserve rich geometric information. Nevertheless, these methods primarily address view transformation from a projection-based perspective, neglecting the influence of environmental factors such as illumination.

## 3 Method

### 3.1 Overview

As illustrated in Fig. 2, our LIAR comprises a SLLIE, 2D encoder, 2D-IGS, 3D-IDP and predictor. The pipeline begins with SLLIE, which takes a nighttime image $X$ as input and outputs an illumination map $I$ along with an enhanced image $\tilde{X}$. The 2D encoder then extracts image feature $F_{\text{img}}$ from $\tilde{X}$.

Guided by the spatial distribution of $\boldsymbol{I}$, 2D-IGS performs adaptive feature sampling on $\boldsymbol{F}_{\text{img}}$, resulting in illumination-aware feature $\boldsymbol{F}_{\text{igs}}$. Subsequently, 3D-IDP integrates $\boldsymbol{I}$ and $\boldsymbol{F}_{\text{igs}}$ to project features into 3D space. Specifically, DepthNet $\Phi_{\text{D}}$ is employed to generate context feature $\boldsymbol{F}_{\text{ctx}}$ and depth prediction $\boldsymbol{D}$, which are processed through BEVPooling to produce context query $\boldsymbol{Q}$. Meanwhile, 3D illumination intensity field $\boldsymbol{S}$ is derived from $\boldsymbol{I}$ to modulate deformable querying between $\boldsymbol{Q}$ and $\boldsymbol{F}_{\text{ctx}}$. Finally, the resulting BEV feature $\boldsymbol{F}_{\text{bev}}$ is passed to the predictor for occupancy prediction.

## 3.2 Selective Low-light Image Enhancement

**Retinex Decomposition.** Based on the Retinex theory [19], given a nighttime image $\boldsymbol{X} \in \mathbb{R}^{3 \times H \times W}$, the Retinex-enhanced image $\boldsymbol{X}_{\text{enh}}$ is formulated as:

$$\boldsymbol{X}_{\text{enh}} = \boldsymbol{X} \oslash \boldsymbol{I}, \quad \boldsymbol{I} \in (0, 1], \tag{1}$$

where $\boldsymbol{I} \in \mathbb{R}^{1 \times H \times W}$ represents the illumination map and $\oslash$ denotes element-wise division. To accurately estimate $\boldsymbol{I}$ from the RGB input, we adapt a stage-wise estimation strategy using a self-calibrated module from SCI [30]. Specifically, $\boldsymbol{I}$ is derived from the input $\boldsymbol{X}$ via a cascaded learning process in a self-supervised manner. To reduce training overhead, we pretrain this module on the nighttime subset of the nuScenes dataset [1] and freeze its weights within the LIAR framework.

**Selective Mechanism.** Leveraging the estimated illumination map $\boldsymbol{I}$, we define an illumination factor $\lambda(\boldsymbol{I})$ to quantify the overall image brightness:

$$\lambda(\boldsymbol{I}) = \frac{1}{|\Omega|} \sum_{(u,v) \in \Omega} \boldsymbol{I}(u, v), \tag{2}$$

where $\Omega$ denotes the set of all pixel coordinates in the image. As illustrated in Fig. 3, we assume that daytime images generally exhibit appropriate illumination and can thus serve as a natural reference for well-exposed scenes. Notably, certain nighttime images yield $\lambda(\boldsymbol{I})$ values comparable to those of daytime images, indicating that not all nighttime images suffer from insufficient illumination. Based on this observation, nighttime images can be categorized into low-light and well-lit subsets. To avoid unnecessary enhancement, only low-light images should be processed, while well-lit ones remain unchanged. To this end, we aim to determine an illumination threshold $t^*$ that effectively separates low-light ($\lambda(\boldsymbol{I}) \leq t^*$) from well-lit ones ($\lambda(\boldsymbol{I}) > t^*$). Following the principle of maximum inter-class variance [33], we formulate the threshold selection as an optimization problem. For a candidate threshold $t$, let $\omega_0(t)$ and $\omega_1(t) = 1 - \omega_0(t)$ denote the proportions of images whose illumination factor satisfies $\lambda(\boldsymbol{I}) \leq t$ and $\lambda(\boldsymbol{I}) > t$, respectively. Let $\mu_0(t)$ and $\mu_1(t)$ denote the corresponding mean illumination values for the two groups. The global mean illumination is computed as:

$$\mu_T = \omega_0(t)\mu_0(t) + \omega_1(t)\mu_1(t), \tag{3}$$

and the inter-class variance is given by:

$$\sigma_b^2(t) = \omega_0(t) \left(\mu_0(t) - \mu_T\right)^2 + \omega_1(t) \left(\mu_1(t) - \mu_T\right)^2. \tag{4}$$

The optimal threshold $t^*$ is obtained by maximizing the inter-class variance:

$$t^* = \arg\max_t \sigma_b^2(t). \tag{5}$$

This threshold separates low-light nighttime images from well-lit ones, enabling selective enhancement while preventing over-processing. The enhanced output $\tilde{\boldsymbol{X}}$ is given by:

$$\tilde{\boldsymbol{X}} = \begin{cases} \boldsymbol{X}_{\text{enh}}, & \text{if } \lambda(\boldsymbol{I}) \leq t^*, \\ \boldsymbol{X}, & \text{otherwise.} \end{cases} \tag{6}$$

## 3.3 2D Illumination-guided Sampling

As illustrated in Fig. 2, the 2D-IGS module enhances 2D image feature representations by learning adaptive sampling points from the illumination maps. Specifically, given an illumination map $\boldsymbol{I}$, we first apply a lightweight guidance network $\Phi_{\text{G}}$ to downsample its resolution, producing $\boldsymbol{I}'$ with the

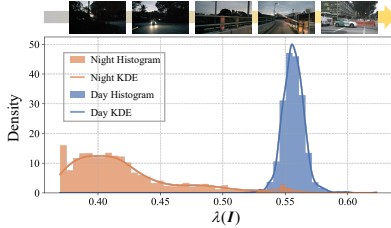

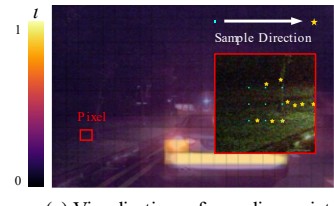

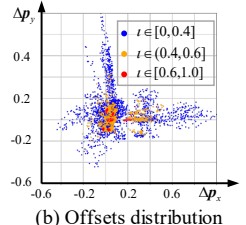

(a) Visualization of sampling points

(b) Offsets distribution

Figure 3: Distributions of the illumination factor in night and day scenes.

Figure 4: (a) Visualization of sampling points in low-light regions. (b) Statistical analysis of the offset distribution.

same resolution as the image feature $\boldsymbol{F}_{\mathrm{img}}$. Subsequently, we compute the illumination guidance map $\boldsymbol{g}$ by applying min-max normalization to the inverse of $\boldsymbol{I}'$:

$$\boldsymbol{g} = \frac{\boldsymbol{I}'^{-1} - \min\left(\boldsymbol{I}'^{-1}\right)}{\max\left(\boldsymbol{I}'^{-1}\right) - \min\left(\boldsymbol{I}'^{-1}\right)}. \tag{7}$$

Next, we map $\boldsymbol{I}'$ to offset $\Delta \boldsymbol{p} = (\Delta \boldsymbol{p}_x, \Delta \boldsymbol{p}_y)$ and weight $\Delta \boldsymbol{w}$ via one convolution layer, allowing illumination-aware control over the receptive field. Then, we modulate $\Delta \boldsymbol{p}$ using $\boldsymbol{g}$:

$$\Delta \tilde{\boldsymbol{p}} = \Delta \boldsymbol{p} \odot \boldsymbol{g}, \tag{8}$$

where $\odot$ denotes element-wise multiplication. Finally, these illumination-aware parameters are then utilized in a deformable convolution [5] operation $\mathcal{F}_{\mathrm{dcn}}(\cdot)$, which performs differentiable warping on the image feature $\boldsymbol{F}_{\mathrm{img}}$. The illumination-aware feature $\boldsymbol{F}_{\mathrm{igs}}$ is computed as:

$$\boldsymbol{F}_{\mathrm{igs}} = \mathcal{F}_{\mathrm{dcn}}(\boldsymbol{F}_{\mathrm{img}}, p + \Delta \tilde{\boldsymbol{p}}) \odot \Delta \boldsymbol{w} + \boldsymbol{F}_{\mathrm{img}}, \tag{9}$$

where $p$ denotes the regular grid of pixel coordinates. Fig.4(a) provides an intuitive visualization of this process, where the sampled points in underexposed regions generate offsets toward adequately-exposed regions. Furthermore, we perform a statistical analysis to investigate the relationship between the offset magnitude at each pixel and its corresponding illumination value, where $\iota$ denotes the pixel value in $\boldsymbol{I}$. As shown in Fig.4(b), pixels in adequately-exposed regions (red and yellow dots) exhibit minor offsets, while those in underexposed regions (blue dots) tend to have significantly larger offsets. These results indicate that 2D-IGS efficiently mitigates feature degradation in 2D representations.

### 3.4 3D Illumination-driven Projection

**Illumination Modeling.** For each BEV grid centered at $(x, y)$, we first uniformly sample $N_z$ points along the vertical axis, yielding heights $\{z_j\}_{j=1}^{N_z}$ within a predefined range. Each 3D point $(x, y, z_j)$ is projected onto the 2D image plane using the camera projection matrix $M \in \mathbb{R}^{3 \times 4}$:

$$d_j \begin{bmatrix} u_j & v_j & 1 \end{bmatrix}^\top = M \begin{bmatrix} x & y & z_j & 1 \end{bmatrix}^\top, \quad \forall j \in \{1, \ldots, N_z\}, \tag{10}$$

where $(u_j, v_j)$ are the projected pixel coordinates, and $d_j$ represents the corresponding depth. The illumination intensity at each height is then sampled from the illumination map $\boldsymbol{I}$ at position $(u_j, v_j)$. Finally, the aggregated illumination value at the BEV location $(x, y)$ is given by:

$$\boldsymbol{S}(x, y) = \frac{1}{N_z} \sum_{j=1}^{N_z} \boldsymbol{I}\left(\lfloor v_j \rfloor, \lfloor u_j \rfloor\right), \tag{11}$$

where $\lfloor \cdot \rfloor$ denotes the floor operation to discretize continuous coordinates into pixel indices. This aggregated $\boldsymbol{S}$ provides an informative estimation of 3D illumination distribution, enabling the model to localize overexposed regions and emphasize them during subsequent feature reasoning.

**BEV Context Refinement.** As shown in Fig. 2, given $\boldsymbol{F}_{\mathrm{igs}}$ as input, the DepthNet $\Phi_{\mathrm{D}}$ is first employed to predict context-aware feature $\boldsymbol{F}_{\mathrm{ctx}}$ and depth $\boldsymbol{D}$. Subsequently, BEVPooling [10] projects $\boldsymbol{F}_{\mathrm{ctx}}$ into BEV space based on $\boldsymbol{D}$, producing context query $\boldsymbol{Q}$. To refine the BEV context features, we propose constructing illumination intensity fields to emphasize overexposed regions. Specifically, each BEV query position $(x, y)$ is first lifted into a vertical set of $N_z$ points $\{z_j\}_{j=1}^{N_z}$

Table 1: Quantitative comparison on the Occ3D-nuScenes dataset. "1f" denotes single-frame method. "2f","4f" and "8f" denotes methods fusing temporal information from 2, 4 and 8 frames.

| Method | mIoU | others | barrier | bicycle | car ↑ | motorcycle | pedestrian | traffic cone | truck | drive. surf. | other flat | sidewalk | terrain | mannade | vegetation | Venue |
|---|---|---|---|---|---|---|---|---|---|---|---|---|---|---|---|---|
| *Train: night    Test: night* | | | | | | | | | | | | | | | | |
| BEVDetOcc (1f) [11] | 13.22 | 0.0 | 0.0 | 0.0 | 34.5 | 0.0 | 0.0 | 0.0 | 0.1 | **58.7** | 1.0 | 24.1 | 26.0 | 19.7 | 21.1 | arXiv22 |
| FlashOcc (1f) [63] | 13.42 | 0.0 | 0.0 | 0.0 | 35.3 | 0.0 | 0.6 | 0.0 | 3.6 | 57.2 | 2.3 | 21.8 | 25.9 | 19.6 | **21.7** | arXiv23 |
| SparseOcc (1f) [27] | 13.32 | 10.4 | 0.5 | 6.4 | 37.3 | 10.9 | 8.2 | 1.4 | 13.1 | 46.4 | 4.4 | 19.7 | 9.6 | 8.8 | 9.3 | ECCV24 |
| OPUS (1f) [40] | 11.32 | 0.2 | 0.0 | 0.0 | 33.2 | 4.6 | 0.1 | 0.0 | 13.0 | 50.9 | 1.0 | 15.0 | 13.5 | 11.4 | 15.8 | NIPS24 |
| LIAR (1f) | **19.27** | 9.7 | 11.2 | 8.5 | 37.7 | 14.8 | 12.2 | 1.1 | 12.5 | 58.7 | 8.5 | 27.2 | 27.2 | 20.1 | 20.6 | NIPS25 |
| BEVDetOcc (2f) [11] | 15.86 | 0.6 | 0.0 | 0.0 | 40.3 | 0.0 | 6.0 | 0.0 | 4.5 | 62.0 | 3.0 | 28.3 | 28.8 | 23.7 | 24.8 | arXiv21 |
| BEVFormer (2f) [25] | 16.57 | 3.6 | 0.0 | 0.0 | 40.3 | 16.1 | 9.9 | 0.0 | 10.1 | 62.1 | 4.8 | 19.9 | 24.4 | 18.8 | 22.2 | ECCV22 |
| FlashOcc (2f) [63] | 18.15 | 5.4 | 0.0 | 0.0 | 41.5 | 9.5 | 10.5 | 0.0 | 17.7 | 63.5 | 1.7 | 26.8 | 28.0 | 25.0 | 24.6 | arXiv23 |
| FBOcc (2f) [24] | 19.79 | 9.3 | 16.3 | 5.4 | 40.0 | 13.6 | 12.5 | 0.1 | 17.6 | 59.4 | 6.8 | 24.5 | 29.7 | 20.0 | 21.9 | CVPR23 |
| OPUS (2f) [40] | 12.77 | 1.5 | 0.0 | 0.0 | 35.4 | 8.5 | 1.1 | 0.0 | 17.0 | 52.2 | 2.2 | 16.6 | 14.6 | 13.0 | 16.1 | NIPS24 |
| SparseOcc (2f) [27] | 14.29 | 12.0 | 1.9 | 7.9 | 37.9 | 9.8 | 9.3 | 0.4 | 16.6 | 46.9 | 4.8 | 21.8 | 11.4 | 9.7 | 9.9 | ECCV24 |
| COTR (2f) [31] | 20.01 | **15.3** | 0.3 | 1.5 | **44.0** | 18.1 | 10.1 | 0.0 | 7.7 | 63.2 | 7.0 | 30.3 | 31.1 | 25.1 | 26.6 | CVPR24 |
| LIAR (2f) | **22.09** | 13.0 | 5.3 | 13.5 | 42.8 | 19.3 | 18.6 | 1.1 | 20.2 | 64.3 | 2.8 | 29.0 | 29.3 | 25.4 | 24.7 | NIPS25 |
| *Train: day & night    Test: night* | | | | | | | | | | | | | | | | |
| BEVDetOcc (1f) [11] | 18.96 | 4.7 | 22.5 | 2.6 | 38.5 | 6.6 | 6.5 | 0.0 | 12.5 | 63.6 | 5.7 | 29.0 | 28.8 | 21.3 | 23.1 | arXiv22 |
| FlashOcc (1f) [63] | 18.93 | 4.3 | 1.0 | 0.0 | 38.7 | 7.7 | 5.8 | 0.0 | 13.6 | 60.6 | 2.1 | 27.1 | 30.0 | 21.7 | 22.5 | arXiv23 |
| LIAR (1f) | 23.67 | 12.0 | 36.1 | 14.5 | 40.7 | 19.2 | 14.0 | 1.3 | 19.3 | 60.5 | 11.1 | 28.9 | 30.5 | 21.4 | 21.8 | NIPS25 |
| BEVDetOcc (2f) [11] | 21.98 | 9.1 | 16.5 | 2.4 | 44.4 | 8.2 | 11.4 | 0.0 | 28.9 | 64.6 | 8.6 | 29.9 | 31.4 | 26.1 | 26.3 | arXiv21 |
| BEVFormer (4f) [25] | 13.77 | 3.1 | 19.8 | 0.7 | 44.1 | 16.6 | 14.5 | 0.0 | 22.3 | 35.8 | 5.6 | 13.9 | 8.4 | 3.1 | 5.0 | ECCV22 |
| FlashOcc (2f) [63] | 23.40 | 13.4 | 18.5 | 2.3 | 46.5 | 10.7 | 14.3 | 0.0 | 30.0 | 66.5 | 5.9 | 32.5 | 32.9 | 28.1 | 26.1 | arXiv23 |
| OPUS (8f) [40] | 20.28 | 14.0 | 10.4 | 12.1 | 40.4 | 15.2 | 13.0 | 0.0 | 27.9 | 62.9 | 5.0 | 25.5 | 19.8 | 17.1 | 20.6 | NIPS24 |
| SparseOcc (8f) [27] | 22.79 | 15.8 | 40.0 | 23.0 | 43.5 | 15.3 | 13.6 | 0.4 | 28.9 | 58.8 | 10.2 | 26.2 | 16.7 | 13.2 | 13.5 | ECCV24 |
| COTR (2f)[31] | 25.17 | 16.0 | 41.5 | 8.9 | 42.1 | 17.4 | 12.5 | 0.0 | 26.6 | 65.2 | 11.2 | 27.0 | 33.7 | 24.8 | 25.6 | CVPR24 |
| LIAR (2f) | **27.33** | 15.9 | 37.7 | 19.0 | 45.4 | 19.1 | 17.8 | 1.9 | 33.6 | 67.1 | 8.1 | 31.2 | 33.7 | 27.4 | 24.7 | NIPS25 |

along the height dimension. For each point $(x, y, z_j)$, its corresponding 2D coordinates are computed using the projection function $\mathcal{P}$ defined in Eq. 10. The residual query $\boldsymbol{Q}'$ at $(x, y)$ is defined as:

$$\boldsymbol{Q}'(x,y) = \sum_{j=1}^{N_z} \mathcal{F}_{\text{dca}}\left(\boldsymbol{Q}(x,y), \mathcal{P}\left(x, y, z_j\right), \boldsymbol{F}_{\text{ctx}}\right). \tag{12}$$

In this formulation, $\mathcal{F}_{\text{dca}}$ denotes the deformable cross-attention module [67], which leverages the query $\boldsymbol{Q}(x,y)$ to sample relevant context feature $\boldsymbol{F}_{\text{ctx}}$ around the projected coordinates. Finally, the refined BEV feature $\boldsymbol{F}_{\text{bev}}$ is given by:

$$\boldsymbol{F}_{\text{bev}} = \boldsymbol{Q} + \boldsymbol{Q}' \odot \boldsymbol{S}, \tag{13}$$

where the illumination intensity field $\boldsymbol{S}$ acts as a spatial weighting factor.

### 3.5 Training Loss

First, we employ a weighted cross-entropy loss $\mathcal{L}_{\text{ce}}$ to supervise the learning of the predictor:

$$\mathcal{L}_{\text{ce}} = -\sum_{v=1}^{N_{\text{vox}}} \sum_{m=1}^{N_{\text{cla}}} c\, \hat{r}_{v,m} \log\left(\frac{e^{r_{v,m}}}{\sum_m e^{r_{v,m}}}\right), \tag{14}$$

where $N_{\text{vox}}$ and $N_{\text{cla}}$ denote the total number of voxels and classes. Here, $r_{v,m}$ is the prediction logit for $v$-th voxel belonging to class $m$, $\hat{r}_{v,m}$ is the corresponding label. The weight $c$ is a class-wise balancing factor computed as the inverse of the class frequency. In addition, inspired by MonoScene [3], we incorporate two auxiliary losses: $\mathcal{L}_{\text{scal}}^{\text{sem}}$ and $\mathcal{L}_{\text{scal}}^{\text{geo}}$, which respectively regularize the semantic and geometric consistency of the predictions. Finally, the total training loss $\mathcal{L}_{\text{t}}$ is formulated as:

$$\mathcal{L}_{\text{t}} = \alpha \mathcal{L}_{\text{ce}} + \beta \mathcal{L}_{\text{scal}}^{\text{sem}} + \gamma \mathcal{L}_{\text{scal}}^{\text{geo}}, \tag{15}$$

where $\alpha$, $\beta$ and $\gamma$ are hyper-parameters and we empirically set to 10, 0.2, and 0.2, respectively.

Table 2: Quantitative comparison on the nuScenes-C dataset under three severity levels.

| Method | mIoU | others | barrier | bicycle | bus | car | const. veh. | motorcycle | pedestrian | traffic cone | trailer | truck | drive. surf. | other flat | sidewalk | terrain | manmade | vegetation |
|---|---|---|---|---|---|---|---|---|---|---|---|---|---|---|---|---|---|---|
| *Severity: easy* | | | | | | | | | | | | | | | | | | |
| BEVDetOcc (1f) [11] | 15.08 | 0.5 | 12.4 | 1.3 | 14.1 | 24.9 | 7.7 | 3.8 | 7.8 | 4.5 | 2.1 | 10.1 | 62.5 | 12.0 | 25.5 | 32.0 | 17.1 | 18.2 |
| FlashOcc (1f) [63] | 12.47 | 0.2 | 7.5 | 1.9 | 16.1 | 19.1 | 6.6 | 3.0 | 5.6 | 3.4 | 4.6 | 9.1 | 49.7 | 9.9 | 19.7 | 25.8 | 14.9 | 14.7 |
| LIAR (1f) | 22.52 | 3.7 | 19.4 | 12.1 | 24.4 | 31.8 | 11.0 | 14.8 | 16.3 | 14.4 | 11.6 | 17.9 | 67.9 | 21.5 | 34.1 | 38.2 | 21.8 | 22.2 |
| BEVDetOcc (2f) [11] | 20.22 | 0.9 | 18.2 | 4.3 | 20.4 | 35.6 | 13.2 | 8.3 | 12.4 | 9.3 | 3.4 | 18.3 | 67.2 | 14.4 | 29.2 | 35.6 | 28.0 | 24.9 |
| BEVFormer (4f) [25] | 14.13 | 0.7 | 20.4 | 3.8 | 28.9 | 33.6 | 4.4 | 8.3 | 11.3 | 7.8 | 10.2 | 18.0 | 34.4 | 14.0 | 17.4 | 11.2 | 7.2 | 8.6 |
| FlashOcc (2f) [63] | 21.83 | 1.7 | 19.5 | 7.7 | 24.0 | 35.0 | 14.6 | 11.6 | 14.3 | 14.3 | 4.4 | 19.0 | 63.3 | 21.3 | 33.2 | 33.3 | 28.9 | 25.1 |
| OPUS (8f) [40] | 23.63 | 4.0 | 22.2 | 14.0 | 31.7 | 37.5 | 16.3 | 15.5 | 13.3 | 13.8 | 12.0 | 24.0 | 65.6 | 23.1 | 34.4 | 29.4 | 21.2 | 23.7 |
| SparseOcc (8f) [27] | 21.98 | 3.9 | 23.0 | 16.6 | 28.7 | 37.5 | 13.2 | 18.6 | 18.9 | 24.7 | 7.4 | 21.8 | 58.0 | 18.9 | 26.9 | 23.3 | 16.5 | 15.8 |
| COTR (2f) [31] | 21.36 | 1.3 | 23.6 | 10.2 | 16.0 | 31.4 | 8.2 | 11.1 | 17.1 | 18.5 | 3.3 | 15.0 | 67.6 | 19.0 | 33.0 | 35.9 | 29.4 | 22.5 |
| LIAR (2f) | 30.66 | 5.1 | 31.6 | 17.2 | 32.2 | 43.8 | 16.7 | 19.8 | 21.5 | 23.9 | 20.0 | 27.6 | 75.5 | 33.1 | 42.5 | 45.0 | 35.7 | 30.1 |
| *Severity: moderate* | | | | | | | | | | | | | | | | | | |
| BEVDetOcc (1f) [11] | 11.50 | 0.3 | 6.6 | 0.8 | 10.0 | 20.7 | 6.1 | 2.8 | 6.0 | 2.2 | 1.0 | 5.5 | 56.5 | 5.7 | 18.2 | 24.8 | 13.6 | 14.8 |
| FlashOcc (1f) [63] | 9.05 | 0.0 | 3.1 | 0.7 | 13.4 | 15.4 | 5.9 | 1.7 | 4.3 | 1.7 | 1.5 | 5.5 | 42.0 | 3.3 | 13.2 | 18.4 | 11.9 | 11.9 |
| LIAR (1f) | 18.44 | 2.2 | 11.1 | 11.1 | 23.0 | 28.0 | 8.5 | 13.8 | 14.0 | 12.5 | 5.0 | 13.7 | 62.8 | 12.0 | 27.5 | 31.4 | 17.8 | 19.0 |
| BEVDetOcc (2f) [11] | 15.95 | 0.4 | 10.3 | 3.2 | 14.9 | 31.8 | 10.1 | 5.7 | 9.7 | 4.9 | 1.7 | 12.6 | 62.4 | 8.5 | 22.2 | 28.8 | 23.6 | 20.5 |
| BEVFormer (4f) [25] | 9.98 | 0.4 | 11.6 | 2.9 | 25.2 | 29.5 | 1.5 | 5.4 | 8.6 | 2.7 | 3.8 | 11.9 | 27.3 | 8.1 | 11.9 | 7.5 | 4.8 | 6.8 |
| FlashOcc (2f) [63] | 17.34 | 0.9 | 12.6 | 5.4 | 17.6 | 31.4 | 10.7 | 8.6 | 11.8 | 9.2 | 2.2 | 12.7 | 60.6 | 14.0 | 26.0 | 28.1 | 23.2 | 19.9 |
| OPUS (8f) [40] | 17.50 | 2.0 | 11.6 | 9.6 | 25.9 | 32.9 | 13.4 | 12.1 | 10.7 | 8.2 | 3.3 | 17.3 | 59.4 | 13.6 | 24.8 | 18.4 | 15.4 | 18.9 |
| SparseOcc (8f) [27] | 16.81 | 1.9 | 13.6 | 11.7 | 22.8 | 33.5 | 9.2 | 15.5 | 16.0 | 19.2 | 2.0 | 16.2 | 53.2 | 8.3 | 19.3 | 16.3 | 13.9 | 13.1 |
| COTR (2f) [31] | 17.77 | 0.8 | 18.2 | 8.6 | 11.4 | 28.4 | 6.8 | 9.7 | 14.1 | 15.0 | 1.3 | 11.3 | 62.9 | 13.2 | 27.1 | 29.9 | 25.1 | 18.3 |
| LIAR (2f) | 25.67 | 2.7 | 20.0 | 13.2 | 28.7 | 39.8 | 12.8 | 17.6 | 18.5 | 20.2 | 11.9 | 22.8 | 72.0 | 25.6 | 37.0 | 39.6 | 29.6 | 24.7 |
| *Severity: hard* | | | | | | | | | | | | | | | | | | |
| BEVDetOcc (1f) [11] | 7.81 | 0.0 | 2.1 | 0.3 | 5.2 | 14.6 | 3.1 | 1.4 | 3.2 | 0.8 | 0.3 | 2.1 | 49.4 | 1.4 | 11.2 | 16.0 | 10.2 | 11.6 |
| FlashOcc (1f) [63] | 5.82 | 0.0 | 0.7 | 0.1 | 7.9 | 11.0 | 1.3 | 0.5 | 2.4 | 0.4 | 0.5 | 2.2 | 34.3 | 0.2 | 7.3 | 11.0 | 9.2 | 9.9 |
| LIAR (1f) | 12.21 | 0.8 | 4.7 | 7.4 | 15.4 | 20.9 | 3.7 | 9.1 | 8.8 | 7.5 | 2.2 | 6.8 | 53.3 | 2.2 | 16.1 | 21.9 | 12.3 | 14.5 |
| BEVDetOcc (2f) [11] | 10.46 | 0.1 | 4.2 | 1.6 | 7.1 | 23.9 | 3.8 | 2.5 | 5.5 | 1.4 | 0.6 | 5.1 | 53.1 | 2.8 | 14.1 | 19.5 | 17.5 | 14.9 |
| BEVFormer (4f) [25] | 5.71 | 0.2 | 5.1 | 0.9 | 13.1 | 22.3 | 0.2 | 2.4 | 4.8 | 0.6 | 0.5 | 5.7 | 19.4 | 2.8 | 6.3 | 5.1 | 3.0 | 4.7 |
| FlashOcc (2f) [63] | 11.76 | 0.2 | 4.8 | 2.0 | 9.7 | 25.0 | 5.0 | 5.2 | 7.2 | 3.9 | 0.8 | 5.3 | 56.3 | 4.9 | 16.5 | 22.3 | 16.4 | 14.3 |
| OPUS (8f) [40] | 10.80 | 0.5 | 3.1 | 5.8 | 14.7 | 25.0 | 10.6 | 5.3 | 6.4 | 2.9 | 0.5 | 8.6 | 49.7 | 3.9 | 13.6 | 10.3 | 9.0 | 13.7 |
| SparseOcc (8f) [27] | 10.77 | 0.8 | 5.1 | 7.9 | 10.8 | 25.9 | 4.2 | 8.2 | 11.3 | 11.0 | 0.3 | 8.3 | 45.2 | 1.0 | 11.6 | 10.6 | 10.9 | 10.0 |
| COTR (2f) [31] | 12.01 | 0.4 | 9.1 | 6.0 | 3.9 | 23.4 | 1.7 | 6.4 | 8.3 | 9.2 | 0.4 | 5.6 | 53.9 | 5.3 | 18.6 | 21.0 | 18.2 | 13.0 |
| LIAR (2f) | 17.31 | 0.8 | 9.1 | 8.3 | 14.2 | 31.5 | 9.0 | 10.2 | 12.3 | 13.2 | 2.3 | 13.1 | 63.0 | 14.4 | 26.0 | 30.1 | 19.5 | 17.6 |

# 4 Experiments

**Dataset.** We evaluate our method on both real and synthetic nighttime scenarios. **(1) Occ3D-nuScenes** [38] includes 700 scenes for training and 150 for validation, with annotations spanning a spatial range of -40m to 40m along both the X and Y axes, and -1m to 5.4m along the Z axis. The occupancy labels are defined using voxels with a size of $0.4m \times 0.4m \times 0.4m$, containing 17 categories. Notably, the training and validation sets include 84 and 15 real-world nighttime scenes, respectively. **(2) nuScenes-C** [51] is a synthetic benchmark that introduces eight types of data corruptions, each applied at three intensity levels to the validation set of nuScenes. Among them, the *Dark* corruption simulates nighttime by reducing brightness and contrast and adding random noise.

**Implementation Details.** We present two variants of LIAR: a non-temporal version and a temporal version that incorporates one historical frame. Both of them are built upon the FlashOcc [63] series and utilize ResNet-50 [7] as the 2D encoder. During training, we employ the AdamW optimizer [29] with a learning rate of $2 \times 10^{-4}$, training each model for 24 epochs. Our implementation is based on MMDetection3D [4], and experiments are conducted on 4 NVIDIA GeForce RTX 4090 GPUs.

## 4.1 Comparison with the State-of-the-Art

To comprehensively evaluate the performance under nighttime conditions, we design two experimental settings. (1) We train the models on the Occ3d-nuScenes dataset using either only nighttime data or the full dataset (daytime + nighttime), and evaluate both models on the nighttime subset (see Tab. 1). (2) We take the model trained on the full dataset and evaluate it on the nuScenes-C (see Tab. 2).

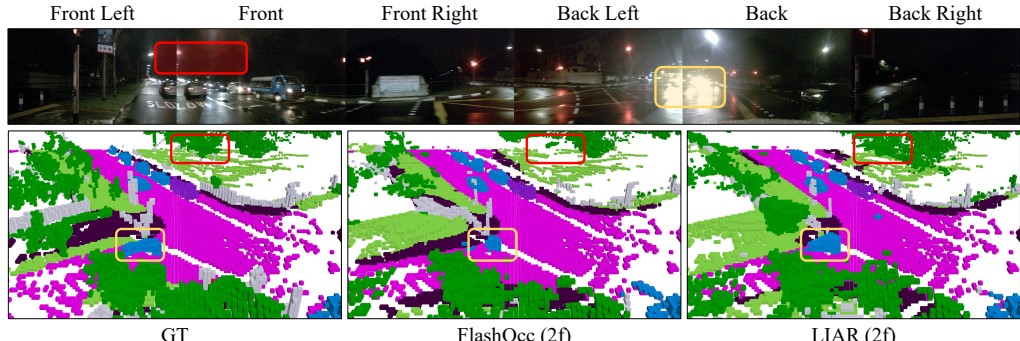

| Front Left | Front | Front Right | Back Left | Back | Back Right |

| GT | FlashOcc (2f) | LIAR (2f) |

Figure 5: Visual results with underexposure and overexposure on the Occ3D-nuScenes dataset.

Table 3: Ablation study of LIAR on the Occ3D-nuScenes.

| LIAR | SLLIE | 2D Feature Enh. | | | View Transformation | | mIoU |
|------|-------|-----|--------|--------|-----------|--------|------|
| | | Add | Concat | 2D-IGS | BEVPooling | 3D-IDP | |
| baseline | | | | | ✓ | | 13.42 |
| i | ✓ | | | | ✓ | | 14.39 |
| ii | | ✓ | | | ✓ | | 13.86 |
| iii | | | ✓ | | ✓ | | 13.81 |
| iv | | | | ✓ | ✓ | | 14.11 |
| v | | | | | | ✓ | 14.88 |
| vi | ✓ | | | ✓ | | ✓ | 15.31 |

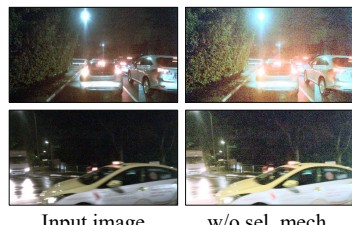

Input image    w/o sel. mech.

Figure 6: Adverse effect of indiscriminate enhancement.

**Comparison on Real-world Dataset.** We first evaluate our LIAR on the Occ3d-nuScenes dataset. As shown in Tab. 1, LIAR consistently achieves superior performance across both training settings. Notably, when trained solely on nighttime data, our model achieves the best performance, outperforming the second-best approach by 5.85 mIoU. Additionally, when a single history frame is introduced, LIAR (2f) surpasses the suboptimal method (COTR (2f) [31]) by 2.08 in mIoU. Furthermore, when training on the full dataset, LIAR continues to demonstrate its superiority. As illustrated in Fig. 5, LIAR (2f) delivers superior results, particularly in challenging regions such as *car* and *vegetation*, where overexposure and poor visibility degrade the performance of FlashOcc (2f) [63].

**Comparison on Synthetic Dataset.** Given the limited availability of real-world nighttime data, we further evaluate our LIAR on the nuScenes-C dataset. As shown in Tab. 2, LIAR outperforms all competing methods across all three severity levels. Under the *easy* setting, LIAR achieves the best performance in both the single-frame and temporal fusion configurations, surpassing the second-best models by 7.44 and 7.03, respectively. Notably, under the *moderate* and *hard* settings, our single-frame model even outperforms methods that incorporate temporal fusion. For instance, under the *hard* setting, LIAR (1f) outperforms BEVFormer (4f) and SparseOcc (8f) by 6.50 and 1.44 mIoU, respectively. Fig. 7 presents visual comparisons across all three severity levels, illustrating that our model maintains strong performance even under extremely low-light conditions.

### 4.2 Ablation Study

All experiments in this section are conducted on the nighttime split of the Occ3D-nuScenes dataset, with all models are trained exclusively using the cross-entropy loss.

**LIAR Designs.** Tab. 3 presents the ablation results of LIAR. The baseline model excludes SLLIE, 2D-IGS, and 3D-IDP, adopting BEVPooling for view transformation. Building on this, LIAR-i introduces SLLIE for low-light image enhancement, resulting in a 0.97 mIoU improvement. To investigate illumination integration for 2D feature enhancement, LIAR-ii and LIAR-iii apply naive fusion methods by directly adding or concatenating illumination maps with extracted 2D image features, yielding marginal gains of 0.44 and 0.39 mIoU, respectively. In contrast, LIAR-iv replaces these naive approaches with the proposed 2D-IGS, boosting performance to 14.11 mIoU and underscoring the benefits of guided sampling. Furthermore, LIAR-v incorporates 3D-IDP to address overexposure in the 3D space, achieving an additional 1.46 mIoU improvement over the baseline. Finally, LIAR-vi

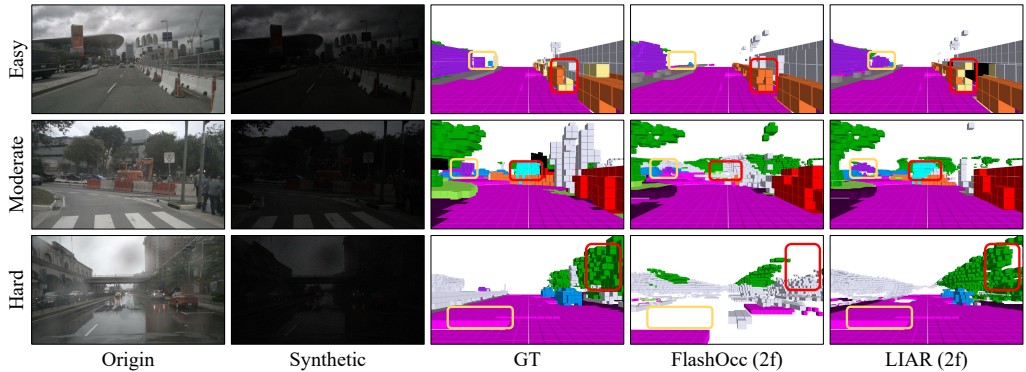

Figure 7: Visual comparisons across the three severity levels on the nuScenes-C dataset.

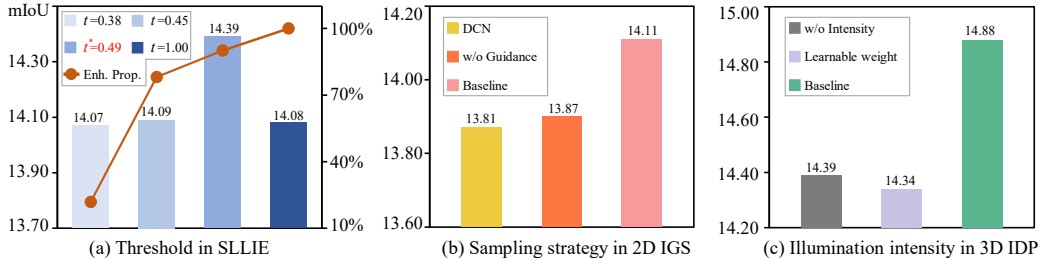

(a) Threshold in SLLIE    (b) Sampling strategy in 2D IGS    (c) Illumination intensity in 3D IDP

Figure 8: Ablation study on the impact of illumination in our SLLIE, 2D-IGS, and 3D-IDP.

integrates SLLIE, 2D-IGS, and 3D-IDP simultaneously, attaining the best overall performance with a 1.89 mIoU gain over the baseline. Overall, each component positively impacts the baseline.

**SLLIE.** Fig. 8(a) presents the ablation study of illumination threshold in SLLIE. The baseline is LIAR-i in Tab. 3. We manually select several illumination thresholds $t$ (0.38, 0.45, 1.00) to compare against the statistically derived optimal $t^*$. The corresponding line graph illustrates how the proportion of enhanced nighttime images varies with different thresholds. As the proportion increases, mIoU also improves, peaking at $t = t^*$. However, when $t = 1.00$, meaning all nighttime images are enhanced indiscriminately, the mIoU decreases from 14.39 to 14.08. This performance drop occurs due to over-brightening images that already possess adequate illumination, which introduces artifacts, amplifies noise, and ultimately deteriorates the overall visual quality, as illustrated in Fig. 6.

**2D-IGS.** Fig. 8(b) shows the ablation study in 2D-IGS. The baseline (pink bar) is LIAR-iv in Tab. 3. When the sampling module is replaced with a standard DCN [5], the mIoU decreases from 14.11 to 13.81, confirming the effectiveness of learning sampling positions according to illumination maps. Additionally, removing only the illumination guidance from the baseline model results in a 0.24 decrease in mIoU, indicating that the guidance map provides essential spatial priors that modulate the sampling offsets. These results validate the rational behind illumination-guided sampling.

**3D-IDP.** Fig. 8(c) illustrates the ablation of illumination intensity field in 3D-IDP. The baseline (green bar) is LIAR-v in Tab. 3. First, removing the illumination intensity field leads to a performance drop from 14.88 to 14.39 mIoU. To further demonstrate the effectiveness of modeling 3D illumination, we design a network that generates BEV weights as a substitute, which results in a decrease of 0.54 mIoU. These results underscore the positive impact of explicitly modeling illumination in 3D space.

## 5    Conclusion

We propose LIAR in this work. Our LIAR is the first occupancy framework that addresses the challenges of nighttime scenes by harnessing illumination-affined representations. LIAR first introduces SLLIE to adaptively enhance limited visibility. Subsequently, 2D-IGS and 3D-IDP are designed to weaken the adverse effects of underexposure and overexposure, respectively. Extensive experiments validate the effectiveness of our LIAR and its superiority in challenging nighttime environments.

**Limitation.** We evaluate our method on both real and synthetic nighttime scenarios using Occ3D-nuScenes [38] and nuScenes-C [50], respectively. For Occ3D-nuScenes, although it serves as the most widely adopted benchmark for occupancy prediction, its nighttime subset is relatively small, and several categories are absent. These limitations may reduce the diversity of evaluated scenarios. As for nuScenes-C, it is a synthetic nighttime dataset generated by darkening daytime images and injecting noise. Due to its synthetic nature, nuScenes-C inevitably differs from real-world data. This domain gap can be broadly categorized into two aspects: First, it retains the semantics and motion patterns of daytime scenes, which differ from actual nighttime driving scenarios. Second, it fails to replicate sensor-induced degradations, such as thermal noise, motion blur, and compression artifacts.

**Broader Impact.** With the increasing development of vision-centric autonomous driving systems, addressing safety concerns under adverse environmental conditions has become critical for ensuring the reliability of autonomous vehicles. LIAR represents a significant step toward robust 3D occupancy prediction in challenging nighttime scenarios. Given the broad applicability of this task, our approach has the potential to benefit a wide range of autonomous driving applications.

## 6   Acknowledgment

This work was supported by the National Science Fund of China under Grant Nos. U24A20330, 62361166670, and 62306155, and by the Technology Major Project No. 2022ZD0116305.

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

# A Technical Appendices and Supplementary Material

Table 4: Comparison of computational cost. All metrics were measured on the NVIDIA 4090 GPU.

| Method | Flops (G) | Memory (GB) | Params (M) | FPS (img/s) | mIoU |
|---|---|---|---|---|---|
| BEVDetOcc [9] | 541.21 | 4.71 | 34.97 | 1.1 | 21.98 |
| FlashOcc [63] | 439.71 | 2.76 | 58.67 | 7.5 | 23.40 |
| OPUS [40] | 215.54 | 2.00 | 73.17 | 22.4 | 20.28 |
| COTR [31] | 740.89 | 12.12 | 37.71 | 0.5 | 25.17 |
| **Our LIAR** (2f) | 690.40 | 5.05 | 64.39 | 4.7 | 27.33 |

Table 5: Quantitative comparison of SLLIE and other low-light image enhancement methods. Here, $\gamma$ denotes the gamma correction factor, and $c, T$ are the clip limit and tile grid size used in CLAHE.

| Method | Description | mIoU |
|---|---|---|
| Gamma Correction [14] | $\gamma = 2.2$ | 13.77 |
| Histogram Equalization [35] | equalize each channel | 13.36 |
| CLAHE [35] | $c = 2.0, T = (8, 8)$ | 13.84 |
| **Our SLLIE** | selective enhancement | 14.39 |

## A.1 Metrics

Following prior works [24, 31, 49, 63], we report the mean Intersection-over-Union (mIoU) as the evaluation metric for voxel-wise occupancy predictions. Notably, in the Occ3D-nuScenes dataset [38], the classes *bus*, *construction vehicle*, and *trailer* are absent from the nighttime validation split. Therefore, the mIoU is calculated over the remaining 14 semantic categories.

## A.2 Computational Cost

Tab. 4 presents the comparison of computational cost on the Occ3D-nuScenes dataset. LIAR achieves the highest mIoU, demonstrating superior performance. However, this improvement comes at the cost of increased computational complexity, primarily due to the Retinex decomposition network used to generate illumination maps. Therefore, future research could focus on more efficient Retinex decomposition techniques that can be seamlessly integrated into the occupancy framework.

## A.3 SLLIE Pretraining

The SLLIE module is pretrained on the nighttime split of the nuScenes training set in a self-supervised manner. Following SCI [30], we use the fidelity loss as follows:

$$\mathcal{L}_f = \sum_{t=1}^{T} \left\| x^t - \left( y + s^{t-1} \right) \right\|^2, \tag{16}$$

where $y$ is the input low-light image, $x^t$ is the is the illumination estimate at stage $t$ and $s^{t-1}$ is the output of the self-calibrated module from the previous stage. When the input image $y$ is already well-exposed (e.g., daytime image), the pseudo-target $y + s^{t-1}$ becomes over-bright, thereby misleading the network to over-enhance the illumination $x^t$. Consequently, $\mathcal{L}_f$ generates erroneous gradients that hinder the model's ability to generalize to genuine low-light scenarios. Given that the full training set comprises a large proportion of daytime images (approximately 88%), jointly training the SLLIE module with the rest of the occupancy model results in optimization collapse. Therefore, we pretrain the SLLIE module on nighttime data and freeze its weights during training to ensure stability.

## A.4 Comparison of SLLIE and Other Methods

We compare the low-light enhancement performance of SLLIE with Gamma Correction [14], Histogram Equalization [35], and CLAHE [35]. All models are trained on the nighttime split of the Occ3D-nuScenes dataset [38], using cross-entropy loss exclusively. As listed in Tab. 5, SLLIE

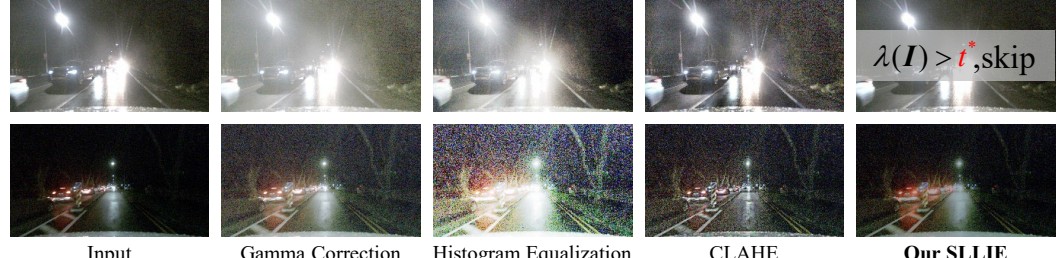

| Input | Gamma Correction | Histogram Equalization | CLAHE | **Our SLLIE** |

Figure 9: Qualitative comparison of SLLIE and other low-light image enhancement methods.

Table 6: Comparison with low-light enhancement methods on the Occ3D-nuScenes.

| Method | mIoU |
|---|---|
| BEVDetOcc [9] | 15.86 |
| BEVDetOcc + LCDPNet [39] | 16.18 (+0.32) |
| BEVDetOcc + CSEC [23] | 16.23 (+0.37) |
| FlashOcc [63] | 18.15 |
| FlashOcc + LCDPNet | 18.29 (+0.14) |
| FlashOcc + CSEC | 18.23 (+0.08) |
| COTR [31] | 20.01 |
| COTR + LCDPNet | 20.22 (+0.21) |
| COTR + CSEC | 20.17 (+0.16) |
| **Our LIAR** | 22.09 |

Table 7: Performance on the Occ3D-nuScenes.

| Method | mIoU | | |
|---|---|---|---|
| | all | day | night |
| BEVDetOcc (2f) [9] | 36.10 | 36.85 | 21.98 |
| BEVFormer (4f) [25] | 23.67 | 24.17 | 13.77 |
| FlashOcc (2f) [63] | 37.84 | 38.90 | 23.40 |
| OPUS (8f) [40] | 33.20 | 33.96 | 20.28 |
| SparseOcc (8f) [27] | 31.08 | 31.59 | 22.79 |
| COTR (2f) [31] | 38.70 | 39.47 | 25.17 |
| **Our LIAR** (2f) | 39.57 | 40.42 | 27.33 |

achieves the best performance. For instance, it surpasses the second best CLAHE [35] by 0.55 mIoU. Fig. 9 presents the qualitative comparisons in challenging lighting conditions. It is evident that other methods tend to amplify overexposed regions and introduce excessive noise in low-light regions, whereas SLLIE effectively avoids over-enhancement and improves overall visual quality.

Moreover, we conduct experiments where we apply state-of-the-art low-light enhancement methods as a preprocessing step before standard occupancy models. Specifically, we use LCDPNet [39] and CSEC [23] as low-light enhancement modules. As shown in Tab. 6, while applying enhancement methods leads to marginal improvements, our LIAR consistently outperforms existing methods. For example, our LIAR outperforms COTR + LCSDPNet and COTR + CSEC by 1.87 and 1.92 mIoU, respectively. This highlights that the improvements are not solely attributable to low-light preprocessing, but also stem from our illumination-aware designs.

## A.5 Performance on Daytime Scenes

While our method is primarily designed for nighttime scenarios, we also evaluate our method on daytime scenes. Tab. 7 shows that our LIAR consistently outperforms existing methods on both the daytime and full datasets, surpassing the second-best COTR [31] by 0.87 and 0.95 mIoU, respectively. These results highlight the generalizability of our approach beyond nighttime conditions.

## A.6 Additional Visual Results

Fig. 10 illustrates results on real-world nighttime data from the Occ3D-nuScenes, while Fig. 11 shows visual comparisons on the same scene under all three severity levels on the nuScenes-C.

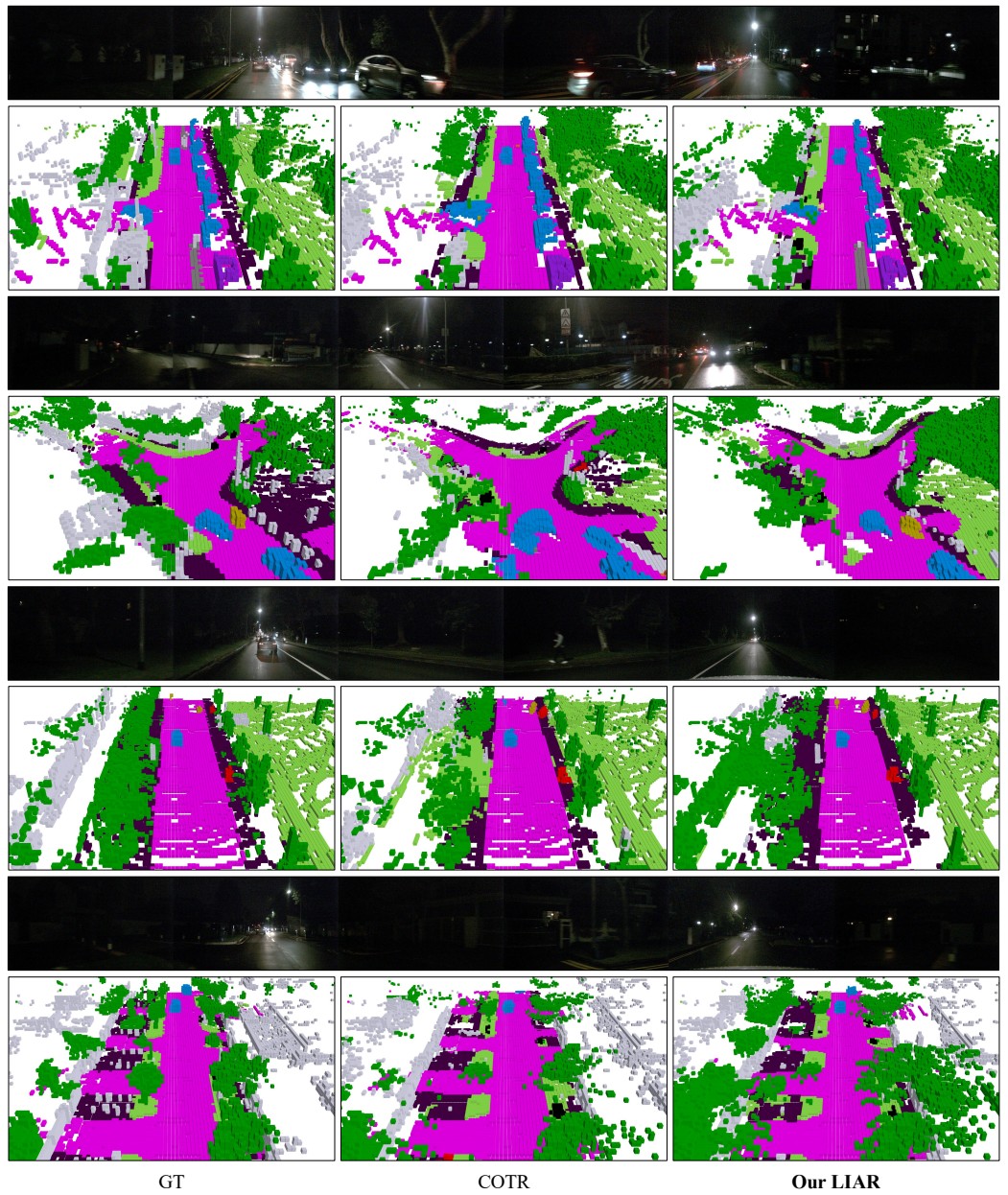

GT       COTR      **Our LIAR**

Figure 10: Qualitative comparison results on the Occ3D-nuScenes dataset.

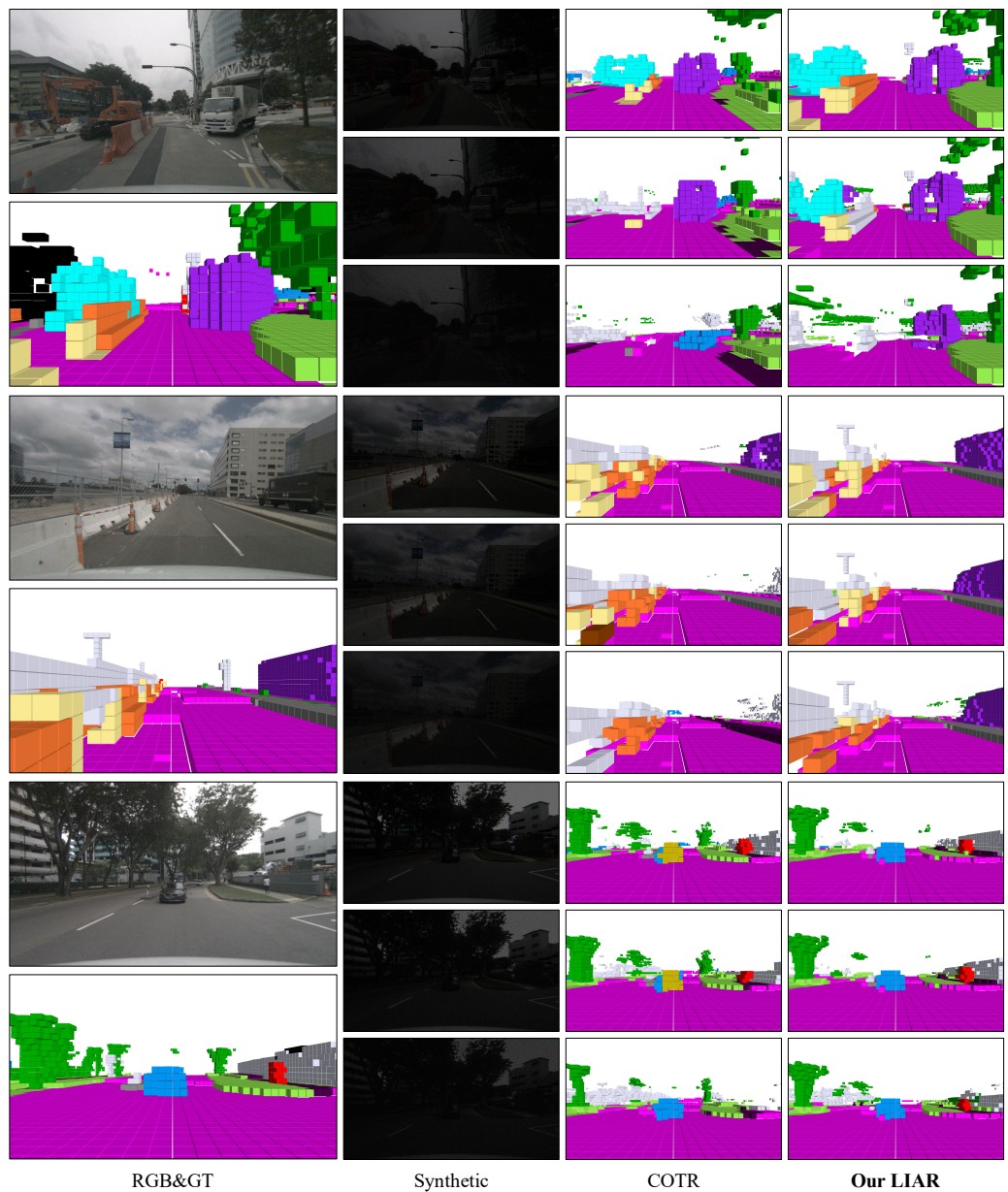

RGB&GT      Synthetic      COTR      **Our LIAR**

Figure 11: Qualitative comparison results on the nuScenes-C dataset across three severity levels.

