# OpenReview forum: "See through the Dark: Learning Illumination-affined Representations for Nighttime Occupancy Prediction"
_NeurIPS.cc/2025/Conference — NeurIPS 2025 poster_

### Official Review · Reviewer_FfZG · 2025-06-26

**Clarity:** 2
**Significance:** 3
**Originality:** 3
**Rating:** 5
**Confidence:** 4

**Summary:**

This paper proposes the LIAR method, which aims to address the occupancy prediction in nighttime scenes. The LIAR method first leverages the SLLIE to enhance the input images adaptively, and then uses 2D-IGS and 3D-IDP to handle under- and over-exposures in 2D and 3D feature space, respectively, before predicting the final occupancies.

**Questions:**

Please refer to the weakness section.

**Ethical Concerns:**

["NO or VERY MINOR ethics concerns only"]

**Final Justification:**

This paper proposes a new method called LIAR for occupancy prediction in nighttime scenes. The proposed LIAR method is demonstrated to be effective for both nighttime and daytime scenes. The rebuttal addresses the concerns of the motivation, method design choices, and generalization ability with discussion and results, which are promised to be included in the revision.

Therefore, I raise my rating.

**Limitations:**

The paper has discussed two limitations: the proposed method may fail under extremely irregular lighting conditions, and the paper is less comprehensively evaluated due to limited data.

While these two points make sense, I think the first point could be elaborated to define better what extremely irregular lighting conditions refer to.

**Quality:**

3

**Strengths And Weaknesses:**

**Strengths.**

+ The paper proposes an important task of occupancy prediction in nighttime scenes, which needs to handle challenging illuminations that may contain both over- and under-exposures.

+ The overall idea make sense to me.

+ The proposed method achieves impressive results on two datasets. Ablations are conducted to help understand the proposed methods.

+ The codes are promised to be released.

Overall, I think this paper proposes to address an important task with a new solution based on a reasonable idea.

**Weaknesses.**

I have some concerns and questions regarding the motivation, method designs and the results.

***Motivation.***

A main argument of this work is that existing low-light image enhancement methods may not handle nighttime well due to the uneven illumination (co-existence of over- and under-exposures). While this point makes senses, the paper misses some relevant works that directly address this issue, for example,

[A] Local Color Distributions Prior for Image Enhancement, ECCV 2022
[B] Region-aware exposure consistency network for mixed exposure correction, AAAI 2024
[C] Color Shift Estimation-and-Correction for Image Enhancement, CVPR 2024

Discussing these methods as baselines is important to validate the proposed method, especially the novelty of SLLIE.



***Method designs.***

(1) The choice of pre-training SCI [29] on the nuScenes dataset [1] lacks justification and convincing evidence (there are only three non-deep learning-based enhancement methods used for comparison in the supplemental).

(2) The illumination threshold seems only consider the average intensity of a whole image, which does not consider local regions. Its effectiveness on images with co-existence of over and under-exposures remains unclear.

(3) The adaptive sampling is interesting but it seems that this assumes well-exposed pixels could be found in the neighborhood of the current under-exposed pixel. This may not be effective for large under-exposed regions. I wonder what is the measurement unit of the shift in Figure 4(b), and whether it would be more helpful if we consider some non-local operations? More discussions and evidences could help strengthen its contribution.

(4) The pipeline contains an implicit order that first addresses underexposures in 2D then over-exposure in 3D. Are there any justifications?

(5) The illumination modeling in 3D-IDP seems a uniform sampling on the illumination map. It is not clear to me why this helps localizes over-exposed regions.

(6) How is the DepthNet \Phi_D trained to make sure depth is accurate? Why is S called illumination field and how is it constructed? Why it can localize the over-exposed regions?




***Results.***

(1) I wonder what the results of daytime scenes are. Given the classifier in SLLIE, the proposed method seems able to handle both daytime and nighttime scenes. Reporting such results could demonstrate its applicability.

(2) How are the competing methods trained? Are they trained on the same data? Have the proposed method compared to baselines that combine image enhancement methods and OP methods?

(3) The paper compares the SLLIE with Gamma correction, Histogram Equalization (HE), and CLAHE (a variant of HE), which are simple non-deep learning-based enhancement methods. Have the proposed method compared with deep learning-based methods?

(4) Have the authors ablated the deformable cross-attention module in 3D-IDP?



There are some minor issues.

(1) Figure 2 is not self-contained: the diagrams of SLLIE, 2D-IGS, and 3D-IDP are quite complex with many symbols and terms (e.g., "modulates feature sampling" and "illumination intensity field") unexplained. The deformable cross-attention module [52] (F_dca) is not shown in Figure 2.

(2) The related work section could be improved.
First, this section could be organized better. While view transformation seems directly related to occupancy prediction, a continuous discussion before switching to low-light image enhancement may be better. Second, discussions on existing occupancy prediction works could be improved. For example, by simply saying MonoScene is a pioneering method, it is not clear what its key idea is and how it relates to this work. Based on this section, view transformation seems an important step for occupancy prediction but it may suffer from uneven illumination. If this is the case, I suggest stating it explicitly.

---

> ### Author Rebuttal · Authors · 2025-07-30
>
> **Motivation weakness 1: Insufficient discussion of recent methods specifically designed for uneven illumination correction.**
>
> Following the suggestion, we incorporate LCDPNet [1] and CSEC [2] as enhancement modules. Please refer to ''Reviewer gNRF - Q5'' and Tab. R3 for detailed results. We will add these results in the revision.
>
> [1] Local Color Distributions Prior for Image Enhancement, ECCV 2022.
>
> [2] Color Shift Estimation-and-Correction for Image Enhancement, CVPR 2024.
>
> **Method design weakness 1:  SLLIE pretraining is under-justified and lacks comparison with deep learning-based enhancement methods.**
>
> **(1)** Please see our response to Reviewer djko (weakness 3). **(2)** In response to your suggestion, Tab. R7 presents a comparison between our SLLIE and both traditional and deep learning-based methods. The baseline refers to the model configuration in the ablation study (see Sec. 4.2, line 219). The results show that deep learning-based methods outperform traditional ones. Nevertheless, our method achieves the highest gain (+0.97 mIoU), highlighting its effectiveness.
>
> Table R7: Comparison of SLLIE with traditional and deep learning-based enhancement methods.
> | Method | mIoU |
> | :---- | :---- |
> | baseline | 13.42 |
> | + Gamma Correction | 13.77 (+0.35) |
> | + Histogram Equalization | 13.36 (-0.06) |
> | + CLAHE | 13.84 (+0.42) |
> | + LCDPNet  | 13.90 (+0.48) |
> | + CSEC  | 13.88 (+0.46) |
> | **+ Our SLLIE** | **14.39 (+0.97)** |
>
> **Method design weakness 2: The illumination threshold overlooks local variations.**
>
> The illumination threshold is a global-level criterion designed to select whether an image broadly suffers from low-light conditions, triggering Retinex-based enhancement via SLLIE. While it does not consider local illumination variations, our framework addresses this through the 2D-IGS and 3D-IDP modules, which are specifically designed to handle under- and over-exposed regions. *This combination ensures both efficient global selection and robust local adaptation across diverse nighttime scenarios.*
>
> **Method design weakness 3: Adaptive sampling may fail in large underexposed regions; clarification on shift units and potential for non-local strategies is needed.**
>
> **(1)** We acknowledge that adaptive sampling may be less effective in extremely large underexposed regions where no well-exposed pixels are available nearby. However, our goal is not to fully reconstruct such severely degraded areas, but to improve local regions where partial illumination is present, which is a more common scenario in real-world nighttime scenes (e.g., streetlights, headlights). **(2)** The unit in Fig. 4 is in pixels. We apologize for the confusion and will clarify this in the revised version. **(3)** As suggested, we conduct experiments by replacing our 2D-IGS module with non-local operations such as DANet [3] and CCNet [4]. For the baseline reported in Tab. R8, we remove SLLIE, 2D-IGS, and 3D-IDP (see Section 4.2, Line 217). While both DANet and CCNet yield performance gains (0.46 and 0.48, respectively), our proposed 2D-IGS still outperforms these methods. We attribute the superior performance of 2D-IGS to its effective use of illumination priors, which guide spatial feature refinement under challenging lighting conditions. Nevertheless, we find the use of non-local operations highly insightful. We will include additional discussion in the revised version. Thanks for your valuable suggestion.
>
> Table R8: Comparison of 2D-IGS with non-local modules.
> | Model | mIoU |
> | :---- | :---- |
> | baseline | 13.42 |
> | +DANet | 13.88 (+0.46) |
> | +CCNet | 13.90 (+0.48) |
> | **+Our 2D-IGS** | **14.11 (+0.69)** |
>
> [3] Dual Attention Network for Scene Segmentation, CVPR 2019.
>
> [4] CCNet: Criss-Cross Attention for Semantic Segmentation, TPAMI 2020.
>
> **Method design weakness 4: The order of addressing underexposure and overexposure requires justification.**
>
> The ordering in our pipeline is intentionally designed based on the following two considerations: **(1)** *Recoverability of information*: Underexposed pixels retain latent details that can be restored using illumination enhancement techniques. In contrast, overexposed (saturated) regions suffer from irreversible information loss due to sensor clipping, making them difficult to recover at the 2D level. **(2)** *Directional flow of the pipeline*: Since occupancy prediction involves transforming features from 2D to 3D, enhancing underexposed regions at the 2D stage helps preserve informative visual cues for subsequent 3D reasoning. Conversely, overexposed regions are better handled in the 3D stage, where spatial and depth context can mitigate the impact of missing information.
>
> **Method design weakness 5: Why the uniform sampling of illumination modeling in 3D-IDP can help localize over-exposed regions?**
>
> **(1)** We would like to clarify that the illumination modeling in 3D-IDP does not perform uniform sampling on the 2D illumination map itself. Instead, it samples points uniformly along the vertical (z) axis in 3D space (see Sec. 3.4). These 3D points are then projected onto the 2D illumination map using camera intrinsics and extrinsics to obtain their corresponding illumination values. **(2)** The reason this helps localize over-exposed regions is that the 2D illumination map inherently reflects pixel-wise brightness, with values normalized between 0 and 1. As a result, the constructed 3D illumination intensity field naturally highlights areas with high illumination values, which correspond to over-exposed regions. This enables the model to identify and focus on those regions during refinement.
>
> **Method design weakness 6: The training of DepthNet and the definition of the illumination field S are unclear, including how $S$ is constructed and how it helps localize over-exposed regions.**
>
> **(1)** The training strategy for DepthNet follows the FlashOcc [5] series. For instance, in our LIAR (2f) setting, the depth predictions are explicitly supervised using ground-truth LiDAR signals. We will include these training details in the revised version for completeness. **(2)** We term the 3D representation of projected illumination intensities as an illumination field $S$, following the convention of naming spatially-varying quantities (e.g., radiance fields, occupancy fields) as “fields” in vision and graphics. **(3)** Please see our response to method design weakness 5 for details.
>
> [5] FlashOcc: Fast and Memory-Efficient Occupancy Prediction via Channel-to-Height Plugin, arXiv 2023.
>
> **Results weakness 1: Results on daytime scenes are not reported.**
>
> Due to space limitations during the rebuttal phase, we kindly refer the reviewer to Tab. R2 in our discussion with Reviewer gNRF.
>
> **Results weakness 2: The compared models lack sufficient details regarding their training protocols and do not incorporate any low-light enhancement techniques.**
>
> **(1)** We adopt two evaluation settings to ensure a fair comparison: (i) training and testing solely on nighttime data, and (ii) training on both daytime and nighttime data while testing on nighttime scenes (see Sec. 4.1, line 195). **(2)** Please see motivation weakness 1 for detail.
>
> **Results weakness 3: Have the proposed SLLIE compared with deep learning-based methods?**
>
> We kindly refer you to method design weakness 1, where we provide a detailed comparison.
>
> **Results weakness 4: Have the authors ablated the deformable cross-attention module in 3D-IDP?**
>
> Yes, and we would like to clarify that our 3D illumination field serves as a spatial weighting mechanism applied on top of the deformable cross-attention outputs. When the deformable cross-attention module is ablated, the model degrades to using standard BEVPooling for view transformation. This corresponds to the baseline (13.42 mIoU) and LIAR-v (14.88 mIoU) in Tab. 3.
>
> **Minor issues 1: Figure 2 is complex and lacks clarity, with unexplained terms and missing components.**
>
> **(1)** The term "modulates feature sampling" refers to the process of generating sampling offsets and weights from the illumination map, which are then used to guide deformable convolution for feature extraction on the image (see Sec. 3.3). The "illumination intensity field" represents the spatial weighting obtained by projecting the 2D illumination map into 3D space (see Sec. 3.4). **(2)** The deformable cross-attention is represented by the red and blue arrows in Fig. 2. Specifically, the red circles indicate the projected points from 3D space to the 2D image plane, while the blue triangles denote the offset sampling locations. The corresponding image features at these locations are aggregated using learned attention weights to produce refined BEV features. Thanks for pointing this out. We will carefully check for similar diagram-text mismatches and revise them accordingly.
>
> **Minor issues 2: Related work section needs better organization.**
>
> **(1)** We will integrate the view transformation part into the occupancy prediction section for coherence. **(2)** We acknowledge the brief description of MonoScene and will include a more detailed discussion of related methods and their relevance in the revision. Thank you for this constructive suggestion.
>
> **Limitations 1: The definition of “irregular lighting conditions” is unclear**.
>
> By “extremely irregular lighting conditions,” we refer to scenes where strong underexposure and overexposure coexist. Examples include transitions between dark tunnels and bright headlights, or flickering lights and wet surface reflections. These challenges hinder accurate illumination estimation and may reduce the effectiveness of our enhancement modules. We appreciate the reviewer’s attention to this detail and will clarify this definition in the revised version.
>
> *We welcome further discussions on the proposed LIAR. The reviewer’s insightful comments are greatly appreciated and will significantly contribute to improving our paper. Thanks!*

---

> > ### Comment · Reviewer_FfZG · 2025-08-04
> >
> > Thank you for your responses, which address my previous concerns well. Up to this point I only have one remaining question.
> >
> > When you say "Conversely, overexposed regions are better handled in the 3D stage, where spatial and depth context can mitigate the impact of missing information", could you elaborate more to explain why spatial and depth information could help address the over-exposure problem?

---

> ### Author Response · Authors · 2025-08-04
>
> Thank you for the follow-up question. The reasons behind are as follows:
>
> **(1) Geometric localization via depth:** Overexposed regions often lack reliable texture or color information in the 2D image space, making them difficult to interpret directly. However, depth information allows us to project these regions into 3D space, providing geometric localization that serves as a foundation for subsequent spatial reasoning.
>
> **(2) Contextual inference through spatial continuity:** Once projected into 3D space, these overexposed regions become spatially anchored within the global scene layout. In such space, scene elements like roads and buildings typically exhibit structural regularities. By leveraging the continuity and consistency of these surrounding structures, the model can infer plausible features for the overexposed areas, thereby compensating for the missing information.
>
> We appreciate your prompt feedback! Please let us know if you have any further questions!

---

> > ### Comment · Reviewer_FfZG · 2025-08-04
> >
> > Thank you for the explanation. It makes sense to me. I will consider raising my rating, but I remain open in the reviewer-AC discussion period.

---

### Official Review · Reviewer_17zV · 2025-07-01

**Clarity:** 3
**Significance:** 2
**Originality:** 3
**Rating:** 4
**Confidence:** 4

**Summary:**

This paper targets the well-recognized challenge of vision-based occupancy prediction in low-light (nighttime) scenarios, where limited visibility and non-uniform illumination degrade perception quality. The authors propose LIAR, a novel framework composed of three key modules:SLLIE (Selective Low-light Image Enhancement); 2D-IGS (2D Illumination-guided Sampling); and3D-IDP (3D Illumination-driven Projection).SLLIE adaptively enhances genuinely dark images using illumination priors derived from daytime scenes. 2D-IGS and 3D-IDP are designed to alleviate underexposure and overexposure effects, respectively, by modifying the feature extraction and projection stages based on spatial illumination. Experimental results on both real-world (nuScenes) and synthetic (nuScenes-C) datasets show that LIAR achieves superior performance over prior state-of-the-art approaches.

**Questions:**

Please see the weaknesses

**Ethical Concerns:**

["NO or VERY MINOR ethics concerns only"]

**Final Justification:**

The paper introduces LIAR, a framework designed to address the challenges of predicting occupancy in nighttime environments.  It is also competitive in daytime environments.   I maintain my score "borderline accept".

**Limitations:**

The authors have discussed the limitations in the last section.

**Quality:**

2

**Strengths And Weaknesses:**

Strengths
1. Novel Illumination-Aware Framework. The paper presents a well-motivated and modular framework that addresses key challenges in nighttime occupancy prediction. The use of illumination maps to guide both 2D feature refinement and 3D projection is conceptually clear and well-executed.
2. Strong Empirical Results. LIAR consistently outperforms state-of-the-art baselines across multiple settings (1f, 2f) and datasets (nuScenes, nuScenes-C). The extensive ablation studies support the importance of each individual module (SLLIE, 2D-IGS, and 3D-IDP).
3. Rich Visualizations. The paper provides compelling visual results, including sampling offset distributions and qualitative comparisons, which effectively convey the intuition behind the proposed modules.

Weaknesses
1. Similarity to Existing Methods (FB-BEV).The proposed BEV context refinement module is architecturally similar to the FB-BEV [1] structure, which also leverages forward and backward BEV projection. However, the paper does not conduct an ablation that isolates this similarity.
Suggestion: Include an experiment or comparison using FB-BEV as the view transformation baseline to help clarify what performance gain is specifically attributable to the 3D-IDP design.
2. Ambiguity in Pretraining Protocol and Fairness Concern.While LIAR is trained on the same data as baselines, the SLLIE module is pretrained to estimate illumination maps from nighttime images. This introduces an implicit supervision signal unavailable to baseline models, potentially giving LIAR an advantage in the training phase. Furthermore, details about this pretraining process are not clearly reported.Please clarify how the Retinex decomposition module is pretrained (e.g., loss, data, duration), and consider reporting performance without SLLIE or using a randomly initialized variant to isolate architectural effectiveness from pretraining benefit.
3. Limited Evaluation Scope and Generalization Concerns.The method is only evaluated on a limited number of real nighttime scenes from nuScenes, which restricts the ability to assess its robustness across diverse conditions.Additionally, the paper does not evaluate whether LIAR degrades performance on daytime scenes, which is critical if the system is expected to work across the full diurnal cycle.Please include performance on the full nuScenes validation set (both day and night scenes) to verify that illumination-aware modules do not harm performance in well-lit conditions. And provide generalization tests to other datasets with nighttime conditions (e.g., OpenScene), or at least conduct cross-dataset evaluation (e.g., train on nuScenes night, test on OpenScene) with quantitative or qualitative results to support generalization claims.

Reference
[1] Li Z, Yu Z, Wang W, et al. FB-BEV: BEV Representation from Forward-Backward View Transformations. ICCV 2023.

---

> ### Author Rebuttal · Authors · 2025-07-30
>
> **Q1: The design of 3D-IDP is similar to FB-BEV and requires an ablation study to verify its contribution.**
>
> **A1: (1)** While our design is inspired by FB-BEV \[1\], our 3D-IDP module introduces a fundamental difference: we construct a 3D illumination intensity field and use it as a spatial weighting mechanism for backward projection. In contrast, FB-BEV aggregates features from forward and backward projections equally, without considering the visibility degradation caused by overexposure under low-light conditions. *This design differs not only in purpose but also in computation: FB-BEV performs symmetric fusion of projections, while our 3D-IDP performs asymmetric, illumination-aware refinement*. **(2)** We provide a comparison with the FB-BEV method in our paper (see Fig. 8(c)), where the gray bar (14.39) corresponds to the view transformation strategy used in FB-BEV, and the green bar (14.88) represents our proposed method. As shown in the ablation results, our design achieves a 0.49 mIoU improvement, further demonstrating the effectiveness of the 3D-IDP module. We sincerely thank the reviewer for this valuable feedback, which helps us better highlight the novelty of our approach. We will elaborate on the differences between 3D-IDP and FB-BEV in the related work section of the revised version.
>
> \[1\] FB-BEV: BEV Representation from Forward-Backward View Transformations, ICCV 2023\.
>
> **Q2: Ambiguity in SLLIE pretraining and fairness concern.**
>
> **A2: (1)** The SLLIE module is pretrained on the nighttime split of the nuScenes training set in a self-supervised manner. We follow the stage-wise illumination estimation with a self-calibrated module structure from SCI \[2\]. We use the fidelity loss: $\\mathcal{L}\_f \= \\sum\_{t=1}^T \\left\\| \\mathrm{x}^t \- \\left( \\mathrm{y} \+ \\mathrm{s}^{t-1} \\right) \\right\\|^2,$ where $\\mathrm{y}$ is the input low-light image, $\\mathrm{x}^t$ is the illumination estimate at stage $t$ and $\\mathrm{s}^{t-1}$ is the output of the self-calibrated module from the previous stage. We train the SLLIE module for 24 epochs using 4 RTX 4090 GPUs, which takes approximately 5 hours. **(2)** As discussed in our response to Reviewer djko (weakness 3), *our SLLIE module cannot be trained end-to-end on the full training set (containing both daytime and nighttime data) due to the nature of its self-supervised loss.* Specifically, the loss formulation leads to incorrect gradient signals when applied to well-lit daytime images, ultimately harming model performance. To address your concern, we compare the performance of models trained with and without SLLIE pretraining on the nighttime subset. As shown in Tab. R6, in the baseline setting where SLLIE, 2D-IGS, and 3D-IDP modules are excluded (see Tab. 3), SLLIE still yields a 0.69 mIoU improvement even when trained from scratch. This highlights that its effectiveness arises not only from pretraining but also from its inherent architectural design.
>
> Table R6: Ablation study of SLLIE.
>
> | SLLIE | Pretrain | mIoU |
> | :---- | :---- | :---- |
> | × | × | 13.42 |
> | √ | × | 14.11(+0.69) |
> | √ | √ | **14.39 (+0.97)** |
>
> \[2\] Toward Fast, Flexible, and Robust Low-Light Image Enhancement, CVPR 2022\.
>
> **Q3: Limited evaluation and generalization concerns.**
>
> **A3: (1)** As discussed with Reviewer gNRF (Q2), we also evaluate our method on the daytime subset and the full dataset. Tab. R2 shows that our model consistently outperforms existing methods without introducing degradation on daytime scenes, demonstrating strong generalization capability. **(2)** We additionally conduct experiments on the synthetic dataset nuScenes-C (see Tab. 2). Given the limited availability of real-world datasets with occupancy labels, we hope the community will contribute more diverse and comprehensive benchmarks in the future. Moreover, as mentioned by Reviewer gNRF, UniScene \[3\], which unifies occupancy and image generation, presents a promising direction for creating high-fidelity synthetic occupancy data for evaluation. We plan to explore this direction further as part of our future work.
>
> Table R2: Quantitative comparison on the Occ3D-nuScenes dataset.
>
> | Model | mIoU (all) | mIoU (night) | mIoU (day) |
> | :---- | :---- | :---- | :---- |
> | BEVDetOcc (2f) | 36.10 | 36.85 | 21.98 |
> | BEVFormer (4f) | 23.67 | 24.17 | 13.77 |
> | FlashOcc (2f) | 37.84 | 38.90 | 23.40 |
> | OPUS (8f) | 33.20 | 33.96 | 20.28 |
> | SparseOcc (8f) | 31.08 | 31.59 | 22.79 |
> | COTR (2f) | 38.70 | 39.47 | 25.17 |
> | **LIAR (2f)** | **39.57** | **40.42** | **27.33** |
>
>  \[3\] UniScene: Unified Occupancy-centric Driving Scene Generation, CVPR 2025\.
>
> *Overall, these insightful suggestions do improve our paper a lot. Please let us know if the reviewer has any further questions, thanks!*

---

> ### Comment · Reviewer_17zV · 2025-08-06
>
> Thank you for the response and explanations.
> Is the result of COTR self-reproduced？ Can results on the Occ3D-nuScenes dataset that are in line with the COTR paper be provided？

---

> > ### Author Response · Authors · 2025-08-06
> >
> > (1) Yes, since no pretrained model weights were released, we reproduced the model using their official code and modified the temporal setting to use 1 historical frame to align with our model's configuration.
> >
> > (2) The results reported in the COTR paper are based on a long-term temporal setting (8 historical frames). However, due to limited GPU resources and time constraints (training would take over 4 days on 4×RTX 4090 GPUs), we are unable to provide results under the original setting during the discussion phase. We appreciate your understanding and will include the results in our revision.
> >
> > Thank you for pointing this out. Please let us know if you have any further questions!

---

### Official Review · Reviewer_djko · 2025-07-02

**Clarity:** 4
**Significance:** 4
**Originality:** 4
**Rating:** 5
**Confidence:** 4

**Summary:**

The paper introduces LIAR, a framework designed to address the challenges of predicting occupancy in nighttime environments. This framework incorporates several innovative components:
•	Selective Low-light Image Enhancement (SLLIE), which uses daytime illumination priors to selectively enhance nighttime images based on their illumination levels.
•	2D Illumination-guided Sampling (2D-IGS) and 3D Illumination-driven Projection (3D-IDP), which help to mitigate the negative effects of underexposure and overexposure in images.
•	The model achieves significant improvements in occupancy prediction in challenging nighttime scenarios by enhancing both 2D and 3D feature representations.

**Questions:**

Suggestions for Improvement:
•	Error Bars and Statistical Significance: Incorporate error bars or statistical significance tests for the experimental results to better evaluate the model's robustness.
•	Dataset Limitations: Discuss the limitations of the datasets used, especially the real-world applicability of synthetic datasets.
•	Broader Impact and Ethical Considerations: Address the broader societal impact of this work, including its potential applications in autonomous vehicles and its ethical implications, such as privacy concerns related to real-time surveillance in nighttime driving scenarios.

**Ethical Concerns:**

["NO or VERY MINOR ethics concerns only"]

**Final Justification:**

The authors has addressed my concerns. This paper can be accepted.

**Limitations:**

see Weaknesses.

**Paper Formatting Concerns:**

No.

**Quality:**

4

**Strengths And Weaknesses:**

Strengths:
1.	Novelty and Originality: The proposed method, LIAR, introduces a new approach to handle the complex problem of nighttime occupancy prediction. The integration of illumination-aware techniques (SLLIE, 2D-IGS, and 3D-IDP) is innovative and targeted towards solving specific issues in nighttime image enhancement and occupancy prediction.
2.	Quality of Experiments: The paper reports extensive experiments on both real and synthetic datasets, showing that LIAR outperforms existing methods, particularly in handling underexposed and overexposed regions. This demonstrates the model’s practical applicability in real-world settings.
3.	Reproducibility: The paper provides sufficient detail on the methodology, including an explanation of the architecture and training process. It also mentions the availability of source code and pretrained models, allowing for reproduction and further research.


Weaknesses:
1.	Limited Error Reporting: The paper does not report statistical error bars, which are critical for understanding the variability and reliability of the results. This could affect the interpretation of how well the model generalizes.
2.	Lack of Broader Context on Dataset Limitations: While the paper discusses the challenges of nighttime scenarios, it does not provide detailed insights into the limitations of the datasets used. For instance, how the synthetic datasets used in testing differ from real-world environments and how the model might perform in other, less controlled settings could be expanded upon.
3.	Over-reliance on Pretrained Models: The framework uses pretrained models in several components (e.g., SLLIE), which could limit the model’s generalization capability to unseen scenarios if the pretrained models are not sufficiently robust across diverse nighttime conditions.

---

> ### Author Rebuttal · Authors · 2025-07-30
>
> **To Weakness 1 and Suggestion 1: Lack of statistical error analysis.**
>
> Due to limited GPU resources and the fact that training on the full dataset takes approximately two days, we currently report results based on the nighttime subset. We conducted five independent runs of our LIAR. As listed in Tab. R5, our model demonstrates consistent performance across multiple runs. We appreciate your emphasis on statistical rigor, which is crucial for ensuring the reliability and reproducibility of research findings. In the revised version, we will extend our statistical analysis to the full dataset, including results with error bars and statistical significance tests.
>
> Table R5: Model performance across independent runs.
>
> | Run | mIoU |
> | :---- | :---- |
> | 1 | 19.27 |
> | 2 | 19.19 |
> | 3 | 19.30 |
> | 4 | 19.26 |
> | 5 | 19.28 |
> | Mean  | 19.26 |
> | Std | 0.04 |
>
> **To Weakness 2 and Suggestion 2:** **Lack of discussion on dataset limitations and the real-world applicability of synthetic datasets.**
>
> **(1)** We evaluate our method on both real and synthetic nighttime scenarios using Occ3D-nuScenes \[1\] and nuScenes-C \[2\], respectively. For Occ3D-nuScenes, although it serves as the most widely adopted benchmark for occupancy prediction, its nighttime subset is relatively small, and some categories are absent under nighttime conditions (see Supp. A.1). Such limitations may reduce the diversity of evaluated scenarios and affect assessment on rare classes. As for nuScenes-C, it is a synthetic nighttime dataset generated by darkening daytime images and injecting noise (see Sec. 4, line 186). Due to its synthetic nature, nuScenes-C inevitably differs from real-world data. This domain gap can be broadly categorized into two aspects: First, it retains the semantics and motion patterns of daytime scenes, which differ from actual nighttime driving scenarios characterized by sparse traffic and high illumination contrast. Second, it fails to replicate sensor-induced degradations, such as thermal noise, motion blur, and compression artifacts, all of which are commonly observed in real low-light conditions. **(2)** We acknowledge the limitations of synthetic data, but it remains highly valuable for training robust models under low-data regimes. Synthetic datasets like nuScenes-C offer scalable generation and controllable diversity, helping to compensate for the scarcity of real nighttime occupancy data. Prior works (e.g., SYNTHIA [3], GTA5 [4], CARLA [5]) have shown that synthetic data can effectively complement real-world datasets and improve model performance. We thank the reviewer for raising this important and thoughtful point. We will incorporate a more thorough discussion of dataset limitations in the revised version, and hope our reflections will encourage further efforts toward building more diverse benchmarks for nighttime occupancy perception.
>
> [1] Occ3D: A Large-Scale 3D Occupancy Prediction Benchmark for Autonomous Driving, NeurIPS 2023\.
> [2] Benchmarking and Improving Bird’s Eye View  Perception Robustness in Autonomous Driving, TPAMI 2025\.
> [3] The SYNTHIA Dataset: A Large Collection of Synthetic Images for Semantic Segmentation of Urban Scenes, CVPR 2016\.
> [4] Playing for Data: Ground Truth from Computer Games, ECCV 2016\.
> [5] CARLA: An Open Urban Driving Simulator, CoRL 2017\.
>
>
> **To Weakness 3: Over-reliance on the pretrained model (SLLIE).**
>
> **(1)** The SLLIE module is pretrained and frozen due to the nature of its self-supervised loss function, which is only meaningful when applied to low-light images. Following SCI \[6\], we use the fidelity loss as follows: $\\mathcal{L}\_f \= \\sum\_{t=1}^T \\left\\| \\mathrm{x}^t \- \\left( \\mathrm{y} \+ \\mathrm{s}^{t-1} \\right) \\right\\|^2,$ where $\\mathrm{y}$ is the input low-light image, $\\mathrm{x}^t$ is the is the illumination estimate at stage $t$ and $\\mathrm{s}^{t-1}$ is the output of the self-calibrated module from the previous stage. When the input image $\\mathrm{y}$ is already well-exposed (e.g., daytime image), the pseudo-target $\\mathrm{y} \+ \\mathrm{s}^{t-1}$ becomes over-bright, thereby misleading the network to over-enhance the illumination $\\mathrm{x}^t$. Consequently, $\\mathcal{L}\_f$ generates erroneous gradients that hinder the model’s ability to generalize to genuine low-light scenarios. Given that the full training set comprises a large proportion of daytime images (approximately 88%), jointly training the SLLIE module with the rest of the occupancy model results in optimization collapse. Therefore, we pretrain the SLLIE module on nighttime data and freeze its weights during training to ensure both stability and effectiveness. **(2)** Tab. R6 compares the performance of models with and without SLLIE pretraining. For the baseline model, we remove SLLIE, 2D-IGS, and 3D-IDP (see Tab. 3). Notably, even when trained jointly without pretraining, SLLIE still yields a performance gain of 0.69 mIoU over the baseline. This demonstrates that the performance gain of SLLIE stems primarily from its architectural design rather than pretraining, as evidenced by the relatively small performance gap between the pretrained and non-pretrained settings.
>
> Table R6: Ablation study of SLLIE.
> | SLLIE | Pretrain | mIoU |
> | :---- | :---- | :---- |
> | × | × | 13.42 |
> | √ | × | 14.11 (+0.69) |
> | √ | √ | **14.39 (+0.97)** |
>
> \[6\] Toward Fast, Flexible, and Robust Low-Light Image Enhancement, CVPR 2022\.
>
> **To Suggestion 3: Address the broader societal impact of this work.**
>
> This work aims to enhance 3D occupancy perception under low-light conditions, offering clear societal benefits for autonomous driving by improving safety, particularly in challenging nighttime environments. More reliable occupancy prediction in such scenarios can help reduce accidents and facilitate the broader deployment of autonomous systems. At the same time, we acknowledge potential ethical considerations, including privacy concerns associated with real-time visual sensing in public and residential areas. We advocate for the responsible use of such technologies, guided by established data protection regulations and best practices such as anonymization, restricted data access, and secure system design.
>
> *We sincerely thank the reviewer for the professional suggestions and kind support. We hope our reply would address some concerns.*

---

### Official Review · Reviewer_gNRF · 2025-07-04

**Clarity:** 3
**Significance:** 3
**Originality:** 2
**Rating:** 3
**Confidence:** 3

**Summary:**

This paper proposes a novel framework for 3D occupancy prediction in challenging nighttime driving scenarios. The method addresses visibility issues caused by underexposure and overexposure through the proposed key components.

**Questions:**

See Weaknesses

**Ethical Concerns:**

["NO or VERY MINOR ethics concerns only"]

**Limitations:**

See Weaknesses

**Quality:**

2

**Strengths And Weaknesses:**

Strengths:
This paper shows clear motivation and well-structured methodology targeting real-world nighttime perception challenges.

Weakness:
The computational cost is insufficiently analyzed. Only limited comparisons of latency and FLOPs are provided in the supplementary material, and only against a few baselines, making it difficult to evaluate the method's efficiency and suitability for real-time deployment.

The paper does not report performance on the full test set or on daytime scenes. While the method is designed for nighttime, it is important to demonstrate that it maintains competitive accuracy during the day to ensure overall robustness.

Visual comparisons are only made against one baseline (FlashOcc), which limits the interpretability and fairness of the qualitative analysis.

The evaluation on real nighttime data is based on a relatively small subset (Occ3D-nuScenes), with most improvements shown on synthetic corruptions. This restricts the strength of real-world claims.

Existing baselines are designed for general or daytime conditions and are evaluated on nighttime data without any low-light enhancement. A more fair comparison would involve combining standard occupancy prediction methods with state-of-the-art low-light enhancement techniques as a preprocessing step. Without such a baseline, it is difficult to disentangle the gains from the proposed architecture versus those from the illumination enhancement alone.

The paper lacks comparison with the latest occupancy prediction methods from recent conferences, which weakens the claim of state-of-the-art performance:
[1] UniScene: Unified Occupancy-centric Driving Scene Generation
[2]  STCOcc: Sparse Spatial-Temporal Cascade Renovation for 3D Occupancy and Scene Flow Prediction

---

> ### Author Rebuttal · Authors · 2025-07-30
>
> **Q1: The computational cost is insufficiently analyzed.**
>
> **A1:** We add comparisons with additional baselines and report detailed metrics, including FLOPs, memory consumption, parameter count, and FPS. As shown in Tab. R1, our model achieves superior performance in terms of accuracy under nighttime conditions. However, it exhibits a gap in FPS and other efficiency metrics compared to several lightweight alternatives. Bridging this gap remains a valuable direction for future work.
>
> Table R1: Comparison of computational cost.
> | Model | FLOPs (G) | Memory (GB) | Params (M) | FPS (img/s) | mIoU |
> | :---- | :---- | :---- | :---- | :---- | :---- |
> | BEVDetOcc  | 541.21 | 4.71 | **34.97** | 1.1 | 21.98 |
> | FlashOcc  | 439.71 | 2.76 | 58.67 | 7.5 | 23.40 |
> | OPUS  | **215.54** | **2.00** | 73.17 | **22.4** | 20.28 |
> | COTR  | 740.89 | 12.12 | 37.71 | 0.5 | 25.17 |
> | **LIAR**  | 690.40 | 5.05 | 64.39 | 4.7 | **27.33** |
>
>
> **Q2: The proposed method is not tested in daytime scenes.**
>
> **A2:** While our method is primarily designed for nighttime scenarios, we fully agree that demonstrating robustness under daytime scenes is important. Tab. R2 shows that our LIAR consistently outperforms existing methods on both the daytime and full datasets, surpassing the second-best COTR by 0.87 and 0.95 mIoU, respectively. These results highlight the generalizability of our approach beyond nighttime conditions. We appreciate the reviewer’s insightful comment and will include the evaluation results in the revised version to provide a more comprehensive assessment.
>
> Table R2: Quantitative comparison on the Occ3D-nuScenes dataset.
> | Model | mIoU (all) | mIoU (day) | mIoU (night) |
> | :---- | :---- | :---- | :---- |
> | BEVDetOcc (2f) | 36.10 | 36.85 | 21.98 |
> | BEVFormer (4f) | 23.67 | 24.17 | 13.77 |
> | FlashOcc (2f) | 37.84 | 38.90 | 23.40 |
> | OPUS (8f) | 33.20 | 33.96 | 20.28 |
> | SparseOcc (8f) | 31.08 | 31.59 | 22.79 |
> | COTR (2f) | 38.70 | 39.47 | 25.17 |
> | **LIAR (2f)** | **39.57** | **40.42** | **27.33** |
>
> **Q3: Visual comparisons only made against the baseline method (FlashOcc).**
>
> **A3:** Beyond FlashOcc, we intend to include additional visual comparisons with the latest state-of-the-art methods, such as COTR. While the rebuttal phase does not allow for figure uploads, these results will be provided in the revised manuscript to ensure a more thorough visual comparison.
>
> **Q4: Real-world evaluation is limited to a small subset, and most gains are demonstrated on synthetic data.**
>
> **A4:** **(1)** *Real-world datasets with occupancy annotations are inherently limited*. Occ3D-nuScenes is the only publicly available dataset that provides annotated nighttime scenes specifically tailored for 3D occupancy prediction. Although the subset is relatively small, it remains the most widely used benchmark. **(2)** *It is worth noting that real nighttime scenes are particularly challenging due to sensor noise and uneven exposure*. These factors lead to suppressed absolute performance across all models and narrower performance margins. Nevertheless, our method consistently outperforms existing approaches under these conditions. **(3)** *We believe that future research would benefit from larger and more diverse real-world nighttime datasets*. We hope that our work can motivate and facilitate the development of more comprehensive benchmarks.
>
> **Q5: The compared methods do not incorporate low-light enhancement.**
>
> **A5:** We conduct experiments where we apply state-of-the-art low-light enhancement methods as a preprocessing step before standard occupancy models. Specifically, following Reviewer FfZG’s suggestion (see motivation weakness 1), we use LCDPNet \[1\] and CSEC \[2\] as low-light enhancement modules. Due to limited GPU resources, all models are trained and evaluated on the nighttime split of Occ3D-nuScenes. As shown in Tab. R3, while applying enhancement methods leads to marginal improvements, our LIAR  consistently outperforms existing methods. For example, our LIAR outperforms COTR \+ LCSDPNet and COTR \+ CSEC by 1.87 and 1.92 mIoU, respectively. This highlights that the improvements are not solely attributable to low-light preprocessing, but also stem from our illumination-aware designs. We appreciate the reviewer’s insightful comment regarding the fairness of our comparisons. We will incorporate the additional experiments in the revised version to strengthen the completeness of our evaluation.
>
> Table R3: Comparisons on the Occ3D-nuScenes dataset. All models fuse temporal information from 2 frames.
> | Model | mIoU |
> | :---- | :---- |
> | BEVDetOcc | 15.86 |
> | BEVDetOcc \+ LCDPNet | 16.18 (+0.32) |
> | BEVDetOcc \+ CSEC | 16.23 (+0.37) |
> | FlashOcc | 18.15 |
> | FlashOcc \+ LCDPNet | 18.29 (+0.14) |
> | FlashOcc \+ CSEC | 18.23 (+0.08) |
> | COTR | 20.01 |
> | COTR  \+ LCDPNet | 20.22 (+0.21) |
> | COTR  \+ CSEC | 20.17 (+0.16) |
> | **LIAR**  | **22.09** |
>
> \[1\] Local Color Distributions Prior for Image Enhancement, ECCV 2022\.
> \[2\] Color Shift Estimation-and-Correction for Image Enhancement, CVPR 2024\.
>
>
> **Q6:** **Lack of comparison with latest works.**
>
> **A6: (1)** We appreciate the reviewer’s suggestion to compare with recent methods such as UniScene \[3\] and STCOcc \[4\]. These two papers were accepted to CVPR 2025 (June 13–15), after the NeurIPS 2025 submission deadline (May 15). At the time of our submission, neither paper had released full details or official code, making fair and reproducible comparisons infeasible. **(2)** Regarding UniScene, we note that it focuses on generative scene modeling from BEV layouts and does not take RGB inputs, thus diverging from the standard occupancy prediction setting targeted in our work. Nonetheless, its ability to jointly generate RGB and occupancy representations under a unified framework is conceptually inspiring, especially in scenarios with limited data. We will explicitly acknowledge this in the limitations section and discuss it as a potential direction for future work. **(3)** For STCOcc, the pretrained weights (trained with camera mask) are not publicly available. Moreover, training the model on the full dataset takes over two days with four RTX 4090 GPUs. Due to resource constraints, we train and evaluate the model on the nighttime subset. Tab. R4 shows that LIAR (2f) achieves competitive performance compared to STCOcc (16f), despite using far fewer temporal frames.
>
> Table R4: Model performance on the nighttime split of Occ3D-nuScenes.
>
> | Model | mIoU |
> | :---- | :---- |
> | STCOcc (16f) | **24.31** |
> | LIAR (2f) | 22.09 |
>
> \[3\] UniScene: Unified Occupancy-centric Driving Scene Generation, CVPR 2025\.
> \[4\] STCOcc: Sparse Spatial-Temporal Cascade Renovation for 3D Occupancy and Scene Flow Prediction, CVPR 2025\.
>
> *Finally, we appreciate the reviewer's valuable and constructive feedback. Please let us know if having any further questions.*

---

> ### Author Response · Authors · 2025-08-07
>
> Dear Reviewer gNRF,
>
> We hope this message finds you well. As the discussion period is nearing its end in about 48 hours, we would like to know whether our reply has addressed your concerns. If there are any remaining questions or points that require further clarification, please let us know.
>
> We greatly value your insights and appreciate the time and effort you have dedicated to reviewing our paper. Thank you once again for your thoughtful feedback.
>
> Best regards,
>
> Authors

---

> ### Comment · Area_Chair_1R4h · 2025-08-08
>
> Dear **Reviewer gNRF**,
>
> Could you please review the rebuttal and confirm whether the initial questions or concerns have been addressed? Your participation in this author-reviewer discussion would be greatly appreciated. Thank you very much for your time and effort.
>
> Best,
>
> AC

---

### Decision · Program_Chairs · 2025-09-17

**Decision:**

Accept (poster)

**Comment:**

This paper received primarily positive reviews, including two "accept" ratings, one "borderline accept" rating, and one "borderline reject" rating.

The main concerns raised in the initial reviews included issues related to the fairness of the evaluation settings, limitations of the comparison baselines and dataset, unclear details regarding the pretraining process and method designs, and a lack of analysis concerning computational costs. In response, the authors provided explanations and conducted extensive experiments to address these concerns in their rebuttal.

Following the rebuttal and subsequent discussions, no additional issues were raised, and most reviewers leaned toward a positive assessment. The area chair agrees with the reviewers' opinions and therefore recommends accepting this paper. The authors would significantly improve their work by incorporating the additional experiments in their revision.